# Inelastic Dynamic Eccentricities in Pushover Analysis Procedure of Multi-Story RC Buildings

**Athanasios Bakalis** 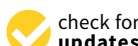**, Triantafyllos Makarios** and **Asimina Athanatopoulou \***

Institute of Structural Analysis and Dynamics of Structures, School of Civil Engineering, Aristotle University of Thessaloniki, GR-54124 Thessaloniki, Greece; abakalis@civil.auth.gr (A.B.); makariostr@civil.auth.gr (T.M.)
\* Correspondence: minak@civil.auth.gr

**Abstract:** A documented pushover procedure on asymmetric, single-story, reinforced concrete (RC) buildings using inelastic dynamic eccentricities is extending in this paper on asymmetric multi-story RC buildings, aiming at the Near Collapse state. The floor lateral static forces of the pushover procedure are applied eccentric to the Mass Centers using appropriate inelastic dynamic or design eccentricities (dynamic plus accidental ones) to safely estimate the ductility demands of both the flexible and stiff sides of the building due to the coupled torsional/translational response. All eccentricities are applied with respect to the "Capable Near Collapse Principal System" of multi-story buildings, which is defined appropriately using the well-known methodology of the torsional optimum axis. Moreover, two patterns of lateral forces are used for performing the analysis, where in the second one an additional top-force is applied to consider the higher-mode effects. A six-story, asymmetric, torsionally-sensitive RC building is examined to verify the proposed pushover procedure relative to the results of non-linear dynamic analysis. The outcomes indicate that the proposed pushover procedure can safely predict the seismic ductility demands at the flexible and stiff sides, providing reliable estimates for the peak inter-story drift-ratios throughout the building as well as a good prediction of the plastic mechanism.

**Keywords:** Capable Near Collapse center of stiffness; inelastic dynamic eccentricities; lateral loading patterns; higher-mode effects; nonlinear static analysis; pushover analysis; response history analysis; torsionally sensitive buildings

## 1. Introduction

The most popular analysis method for the seismic assessment of building structures used in recent years by civil engineers is the non-linear static (pushover) analysis method. For the documented application of pushover analysis on asymmetric multi-story buildings the following must be rationally considered within the linear and nonlinear area of response: (a) the coupled torsional/translational effects, (b) the higher-mode effects, and (c) the P-D effects. In the framework of conventional pushover procedures adopted by modern seismic codes, i.e., Eurocode EN 1998-1 [1] and EN 1998-3 [2], the patterns of floor lateral static loads (in elevation) commonly used are proportional to the fundamental mode-shape or to the inverted-triangular shape or to the uniform shape. Moreover, the lateral static loads are applied on each floor (in-plan) at a shifted position from the Mass Center (CM) by the accidental eccentricity, which is usually equal to 5% of the plan length orthogonal to the direction of excitation. Additionally, there is not a clear identification of the appropriate (principal) building directions which codes propose for the application of horizontal seismic forces to perform push over analysis. Another point related to this method is that codes suggest super-position of non-linear analyses results to consider the spatial seismic action effects. However, this fact is mathematically forbitten in general in the non-linear area. On this point, some other authors [3] suggest a combined load profile acting simultaneously along the two horizontal principal directions of the building, with

base shear proportion to 1:0.3 and 0.3:1 for each main direction. The drawbacks of the pushover procedures proposed by codes are further increased by the absence of the real inelastic center of stiffness and of the definition of inelastic torsional radii in multi-story buildings in the non-linear area. These properties limit the documented definition of the torsional sensitivity of multi-story buildings since, as observed recently in single-story RC buildings [4], torsionally non-sensitive buildings in the linear area can behave sometimes as torsionally sensitive ones during their nonlinear response. All the above mentioned often result in an underestimation of the seismic ductility demands, mainly at the stiff sides and sometimes also at the flexible sides of multi-story buildings, especially in torsionally sensitive ones. Additionally, an underestimation of the seismic ductility demands is often observed at the higher floors of medium or high-rise buildings. So, we see that conventional pushover procedures often lead to an uncertain estimation of the floor inelastic angular deformation demands (also known as inter-story drift ratios or floor drift rotations), which are one of the critical parameters for the seismic ductility demands and the structural damage. All the previous mentioned have already been recognized in single-story buildings [4–8] and in multi-story ones [9]. It is noticed that in the recent release of the Italian Building Code (NTC18) [10], new provisions were inserted about the load profiles and the choice of control nodes in pushover analysis that lead to different capacity curves and to different conclusions of the seismic assessment procedure. These issues were also investigated by other authors [3,11].

In the last two decades, the international scientific community has proposed various pushover procedures to address the major drawbacks abovementioned. These can be categorized into two major types: non-adaptive and adaptive ones. Pushover procedures that belong to the first type use an invariant load vector throughout the analysis while in the second type pushovers the load vector is successively updated in every step of analysis where structural yield is observed or in few ones. The loading vector in most pushover procedures consists of monotonic increasing forces/moments but there are also some procedures that use imposed displacements. The pushover procedures of the first type focus on the contribution of torsional/higher modes to address the irregularity in plan/elevation. Indicatively we mention the modal pushover procedures [12–17], pushovers procedures combined with some kind of dynamic spectrum analysis [18–23], pushover procedures which use inelastic dynamic eccentricities [4–8], or corrective eccentricities [24–26] and direct displacement-based pushover procedures with seismic enforced-displacements [9]. The pushover procedures of the second type mainly focus on the progressive damage of the building and its impact on the dynamic response characteristics due to stiffness degradation in the non-linear area [27–31].

Notwithstanding the wide and often complicated variety of proposed pushover procedures, the scientific community has not yet reached any concrete conclusions. For this reason, the various seismic codes do not directly suggest the use of any specific procedure. This trend has also affected the non-linear analysis software development that has not yet embraced the state-of-the-art. Moreover, the application complexity of some of the proposed procedures is an additional disadvantage that renders them unclear and non-supervisory seismic assessment tools, sometimes more difficult to apply than the Non-Linear Response History Analysis (N-LRHA) which is the benchmark method for the estimation of the seismic demands. So, there are enough grounds for improving and extending existing pushover procedures or for developing new ones with focus on application simplicity and effectiveness of the seismic assessment procedure.

In the current paper, a recently proposed pushover procedure on in-plan irregular single-story RC buildings using inelastic dynamic eccentricities [4,5] is extended to asymmetric multi-story RC buildings. Appropriate inelastic dynamic eccentricities have been proposed produced by an extended parametric analysis on single-story RC buildings (by performing N-LRHA) in the framework of the doctoral dissertation of the first author, aiming directly at the Near Collapse (NC) state of the building. Two inelastic dynamic eccentricities are used that have been calibrated for the safe prediction of the seismic

ductility demands at the building's stiff and flexible sides. If the accidental eccentricity is also considered in analysis, then the inelastic design eccentricities are used. The latter combine the inelastic dynamic eccentricities with the accidental ones in the most unfavorable way. According to the proposed procedure, in order to consider the coupled torsional/translational response, the floor lateral static forces are applying eccentrically to CM, using the inelastic dynamic or design eccentricities, at two in-plan positions: the first one towards the stiff side and the second one towards the flexible side of the building, along each principal direction. The "inelastic dynamic eccentricities" pushover procedure refers to the "Capable Near Collapse Principal System, $CR_{sec}(III_{sec})$, $I_{sec}$, $II_{sec}$" of the single-story building, where $CR_{sec}$ is the Center of Rigidity of the single-story building by assuming the global use of the secant stiffness $EI_{sec}$ at yield in all structural members, $III_{sec}$ is the vertical principal axis passing through $CR_{sec}$ and $I_{sec}$, $II_{sec}$ are the horizontal principal axes [4,5].

For the documented application of the proposed procedure of pushover analysis using inelastic dynamic eccentricities on multi-story RC buildings, aiming at the NC state, the following adjustments must be carried out: (a) the determination of the "Capable Near Collapse Principal System, $CR_{sec}(III_{sec})$, $I_{sec}$, $II_{sec}$" of multi-story buildings using the well documented procedure for the determination of the torsional optimum axis, (b) the determination of the corresponding "Capable Near Collapse Torsional Radii, $(r_{I,sec}, r_{II,sec})$" of multi-story buildings, and (c) the use of two patterns for the floor lateral static forces, which both are modal ones but in the second one an additional lateral force is applied on the top floor of the building. With the previous reformation for the verification of the proposed procedure on multi-story RC buildings at the NC state, the ideal inelastic principal system of the multi-story buildings is defined, which is used as a reference system for the application of the proposed pushover procedure. Additionally, the process of checking the torsional sensitivity of multi-story buildings is facilitated, and the higher-mode effects are rationally considered. The abovementioned (a) to (c) have already been examined in a work of the first two Authors about a proposed pushover procedure with enforced-displacements [9]. The only drawback of the pushover method proposed by EN 1998-1 that remains is the application of superposition techniques (e.g., Square Root of Sum of Squares—SRSS—combination rule) on the separate pushover analyses effects along the two main (principal) orthogonal axes. The proposed pushover methodology on asymmetric multi-story RC buildings aims directly at the safe estimation of the seismic demands at the NC state in terms of floor inelastic angular deformations and displacements, providing that the building under examination shows sufficient ductility and is regular in elevation according to seismic codes. In any case, if the building under examination is not enough ductile or has an irregular layout in elevation, the force-based proposed pushover procedure will highlight all the structural deficiencies as well as the possible plastic mechanisms.

## 2. Methodology

In this section, the methodology used in the recently proposed pushover procedure on single-story RC buildings using inelastic dynamic eccentricities [4] is presented in short and is supplemented by the appropriate adjustments in order to extend it to multi-story RC buildings [9]. The main steps to perform the new proposed pushover procedure on multi-story RC buildings and the key findings of the current investigation are analyzed in the following.

### 2.1. Non-Linear Model

To take rationally into account the Near Collapse (NC) limit state, it is assumed that plastic hinges have been developed at both end-sections of all RC structural elements of the building (full plastic mechanism). In other words, it is considered that all RC end-sections have yielded, and the non-linear model of the building should simulate this ideal state. To accomplish this, all structural elements are supplied with their secant moments of

inertia $I_{sec}$ at yield and the lateral secant stiffness $K_{sec}$ of the building (as a whole) that leads directly to the yield point is represented by the slope of the first branch of an elastoplastic Force-Displacement diagram. This ideal state of full plastic mechanism of the building is called the "Capable Near Collapse" state. It is noted that, according to EN 1998-3, the secant stiffness $EI_{sec}$ at yield is mandatory for all the structural members of the nonlinear model (Informational Annex A, section A.3.2.4(5)). In reality, the Near Collapse state of buildings happens before the development of a full plastic mechanism. However, by considering this "Capable Near Collapse" state, we estimate larger displacements and deformations, because the building is more flexible. Consequently, if it computationally turns out that all structural elements have adequate deformation capacity to resist these displacements and deformations without failure then the response effects caused by a ground motion have been safely controlled. The simulation of the possible inelastic behavior of the element end-sections can be achieved through the insertion of point plastic hinges of Fiber or *P-M*2-*M*3 or *M*3 type.

The secant stiffness $EI_{sec}$ at yield of an RC end-section (corresponding to the entire shear span of that section) is determined according to geometric relationships of the elasticity theory (Equation (1)) with the aid of the semi-empirical models for the chord rotation $\theta_y$ at yield proposed by EN 1998-3 for beam/columns and walls (Equations (2) and (3), respectively), assuming in most cases that the shear span is constant throughout the analysis and equal to half of the clear length of each structural element. Additionally, the ultimate chord rotation $\theta_{um}$ (Equation (5)) can be calculated through plane-section analysis using the material non-linear σ-ε laws and the conventional model for the plastic hinge length $L_{pl}$ (Equation (4)) proposed by EN 1998-3:

$$EI_{sec} = M_y \cdot L_v / 3\theta_y \tag{1}$$

$$\theta_y = \varphi_y \frac{L_v + \alpha_v \cdot z}{3} + 0.0013\left(1 + 1.5\frac{h}{L_v}\right) + 0.13\varphi_y \frac{d_{bL} \cdot f_{sy,m}}{\sqrt{f_{c,m}}} \tag{2}$$

$$\theta_y = \varphi_y \frac{L_v + \alpha_v \cdot z}{3} + 0.002\left(1 - 0.125\frac{L_v}{h}\right) + 0.13\varphi_y \frac{d_{bL} \cdot f_{sy,m}}{\sqrt{f_{c,m}}} \tag{3}$$

$$L_{pl} = \frac{L_v}{30} + 0.2 \cdot h + 0.11 \cdot \frac{d_{bL} \cdot f_{sy,m}}{\sqrt{f_{c,m}}} \tag{4}$$

$$\theta_{um} = \frac{1}{\gamma_{el}}(\theta_y + (\varphi_u - \varphi_y)L_{pl}\left(1 - \frac{0.5L_{pl}}{L_v}\right) \tag{5}$$

where $M_y$ and $L_v$ are the yield moment and the shear span, respectively, $\varphi_y$ is the curvature at yield, $\alpha_v \cdot z$ is the tension shift of the bending moment diagram due to inclined cracking where z is the length of the internal lever arm and $\alpha_v = 1$ when shear cracking is expected, $h$ is the depth of cross-section normal to the yield Moment vector, $d_{bL}$, $f_{sy,m}$, $f_{c,m}$ are the mean tension reinforcement diameter, the mean yield stress of the steel reinforcement and the mean compressive stress of concrete, respectively, and $\varphi_u$ is the ultimate curvature while $\gamma_{el} = 1.7$ is a conversion factor from mean values to mean-minus-one-standard-deviation ones. The yield characteristics of an RC section ($\varphi_y$, $M_y$) and the ultimate ones ($\varphi_u$, $M_u$) can be computed by a plane-section analysis using as parameters the section geometry, the longitudinal and transverse steel reinforcement layout, the constitutive σ-ε relationships of unconfined/confined concrete and of steel reinforcement, and the axial load, usually taken equal to the value used into the seismic combination ($G + \psi_E Q$, where $G$ is the dead load, $Q$ is the live load and $\psi_E$ is the combination coefficient of the live loads which in EN 1998-1 is taken equal to 0.3 for ordinary buildings). Then, considering that the plastic hinge is located exactly at the face of an extreme RC section, the plastic chord rotation is approximately equal to $\theta_{pl,m} = 1/\gamma_{el} \cdot (\varphi_u - \varphi_y)L_{pl}$. Finally, the secant stiffness at yield of each structural element is calculated as the numerical average of the $EI_{sec}$ values of its

two extreme element cross-sections, for positive and negative bending, and is considered constant for the entire member length.

*2.2. Definition of the "Capable Near Collapse Principal System" and of the "Capable Near Collapse Torsional Radii" of Multi-Story Buildings*

In single-story buildings, the abovementioned non-linear model was considered as the most appropriate one for calculations at the NC state [4,5]. The position of the "inelastic" center of stiffness in the floor-diaphragm (corresponding to the "Capable Near Collapse" state) is determined by linear analysis but using the secant stiffness $EI_{sec}$ at yield of the structural elements. The latter is called as the "Capable Near Collapse Center of Stiffness, $(CR_{sec})$" of the single-story building. The horizontal ideal "inelastic" principal axes and the "inelastic" torsional radii of the single-story building are also determined by linear analyses and are called as "Capable Near Collapse Principal Axes, $(I_{sec}, II_{sec})$" and "Capable Near Collapse Torsional Radii, $(r_{I,sec}, r_{II,sec})$", respectively.

For multi-story buildings [9], the corresponding computations are performed using the well-known concept of the torsional optimum axis, which is a vertical fictitious elastic axis around which the mean sum of the squared floor-diaphragm rotations in elevation is minimized [32–34]. According to this methodology, the "Capable Near Collapse Principal System, $CR_{sec}(III_{sec}), I_{sec}, II_{sec}$" of the multi-story building is determined at the building level that is closest to $0.8H_{tot}$ from the base of the building by means of three temporary linear analyses, where $H_{tot}$ is the building height. In short, three loading vectors are used that are proportional to the inverted triangular shape, a vector of floor torques $\mathbf{M}_z$ around $z$-axis for the first analysis and two vectors of floor lateral forces $\mathbf{V}_x$ and $\mathbf{V}_y$ along x and $y$-axis, respectively, for the second and third analyses. The elements of these three vectors have equal values. From the first analysis, the in-plan position of $CR_{sec}$ is determined. Next, the floor lateral forces $\mathbf{V}_x$ and $\mathbf{V}_y$ are applied on $CR_{sec}$ to determine the orientation of the axes $I_{sec}, II_{sec}$ relative to the x, y-axes. The process of performing the elastic analyses is illustrated in Figure 1a–c.

The in-plan position of $CR_{sec}$ (intersection of the vertical inelastic axis $III_{sec}$ with the floor-diaphragm closest to $0.8H_{tot}$) as well as the orientation of the horizontal axes $I_{sec}$ and $II_{sec}$ (Figure 1d) are calculated by Equations (6) and (7), respectively, using the displacement results of the first three analyses:

$$x_c = -\frac{u_{y,M}}{r_{z,M}} \,,\; y_c = +\frac{u_{x,M}}{r_{z,M}} \tag{6}$$

$$\tan(2\hat{\omega}) = \frac{2\, u_{CR,y,Vx}}{u_{CR,x,Vx} - u_{CR,y,Vy}} \tag{7}$$

Therefore, by assuming the global use of the secant stiffness $EI_{sec}$ at yield in all structural members, an ideal 3D "inelastic principal" reference system $CR_{sec}(I_{sec}, II_{sec}, III_{sec})$ of the multi-story building is approximately defined (Figure 1d). This is the "Capable Near Collapse Principal System" of the multi-story building, where (a) its origin is the inelastic center of stiffness $CR_{sec}$ of the multi-story building, (b) the vertical inelastic principal axis $III_{sec}$ is the torsional optimum axis of the multi-story building (Figure 1a) that passes through $CR_{sec}$, and (c) $I_{sec}$ and $II_{sec}$ are the two orthogonal horizontal inelastic principal axes of the multi-story building. The distance from $CR_{sec}$ to CM along the $I_{sec}$ and $II_{sec}$ axes is the inelastic static eccentricity ($e_{R,Isec}$ or $e_{R,IIsec}$, respectively), which is strength dependent.

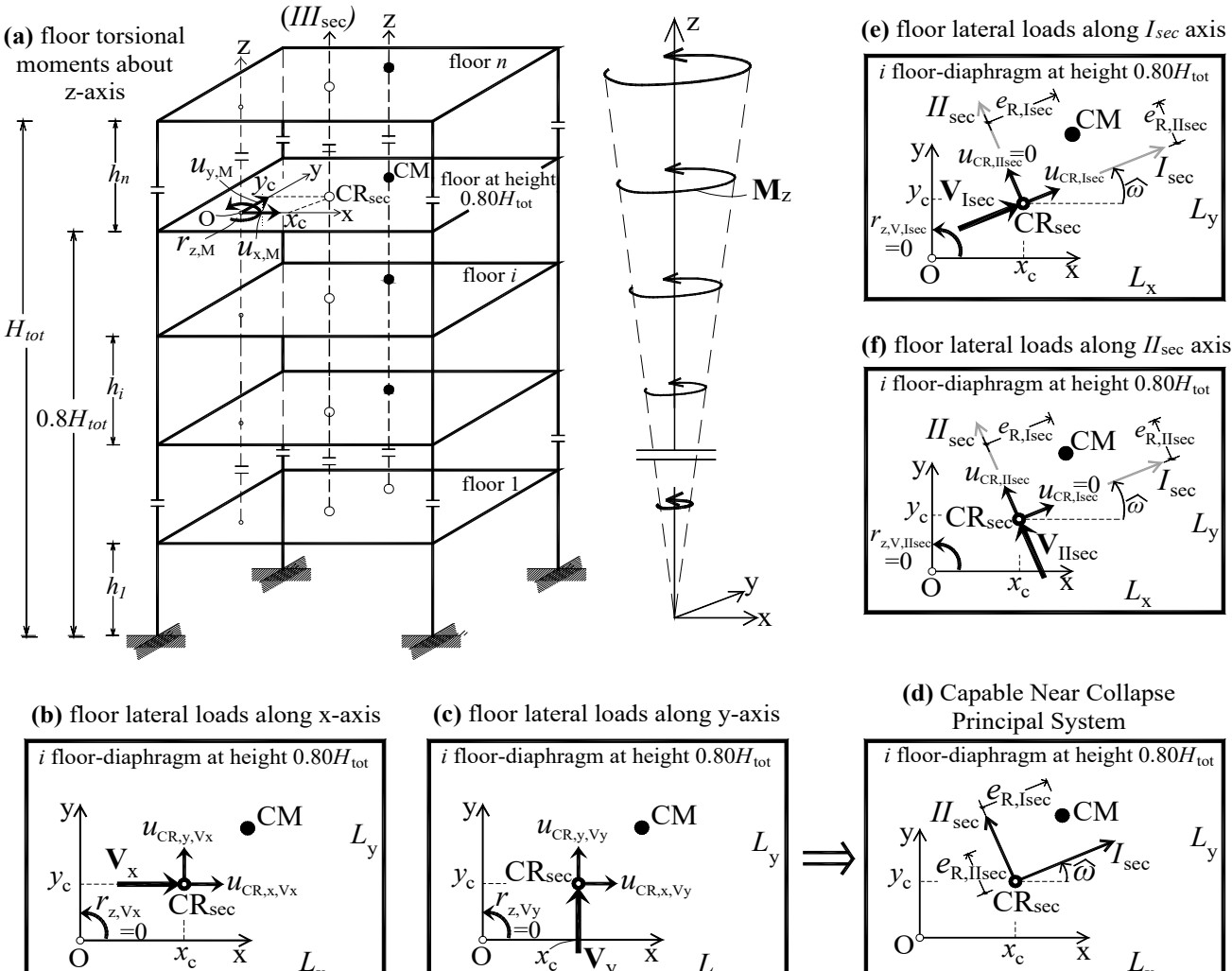

**Figure 1.** Three elastic analyses (**a**–**c**) for the determination of (**d**) "Capable Near Collapse Principal System $CR_{sec}(III_{sec}), I_{sec}, II_{sec}$" and two more elastic analyses (**e**,**f**) for the determination of "Capable Near Collapse torsional radii $r_{I,sec}, r_{II,sec}$", in the non-linear model ($EI_{sec}$) of the multi-story RC building.

Next, the "Capable Near Collapse Torsional Radii ($r_{I,sec}, r_{II,sec}$)" of the multi-story building are calculated by Equation (8), also at the same building level that is closest to $0.8H_{tot}$, by using the displacement results of two more temporary linear analyses with lateral load vectors $\mathbf{V}_{Isec}$ and $\mathbf{V}_{IIsec}$ (equal to $\mathbf{V}_x$ and $\mathbf{V}_y$) applied at the in-plan position of the vertical inelastic torsional optimum axis $III_{sec}(CR_{sec})$ along the $I_{sec}$ and $II_{sec}$ axis, respectively, as illustrated in Figure 1e,f:

$$r_{I,sec} = \sqrt{\frac{u_{CR,IIsec}}{r_{z,M}}} \; , \; r_{II,sec} = \sqrt{\frac{u_{CR,I\,sec}}{r_{z,M}}} \tag{8}$$

The two (mean) values of the inelastic torsional radii $r_{I,sec}, r_{II,sec}$ and the radius of gyration $r_m$ at the building level closest to $0.8H_{tot}$ are used in order to verify the torsional sensitivity of the multi-story. The building is classified as torsionally sensitive when Equation (9) is true [4,9]:

$$r_{I,sec} \text{ or } r_{II,sec} \leq 1.10 \cdot r_m \tag{9}$$

where $r_m = \sqrt{J_m/m}$, $J_m$ is the mass moment of inertia of the floor about a vertical axis passing through its geometric center and $m$ is the mass of the floor closest to the $0.8H_{tot}$ level. The torsional verification can also be performed equivalently by examining the ratio

of the uncoupled fundamental torsional to translational frequencies along each principal direction. The limit value 1.10 shown in Equation (9) is higher than the corresponding limit of 1 used in linear area [35], to consider the increased torsional sensitivity observed in several cases of single-story buildings in the non-linear area, initially characterized as torsionally non-sensitive (in the linear area).

### 2.3. Definition of Inelastic Dynamic Eccentricities for the Safe Prediction of the Ductility Demands at the Stiff and Flexible Sides (in-Plan Irregularity)

In order to predict with safety the floor displacements and the floor angular deformations at the stiff and flexible sides of the multi-story RC buildings along an examined principal direction $I_{sec}$ or $II_{sec}$, the lateral static forces must be applied at two distinct positions in each floor-diaphragm by using the inelastic dynamic eccentricities $e_{stiff}$ and $e_{flex}$ normal to the examined principal direction $I_{sec}$ or $II_{sec}$. The first position is towards the stiff side while the second one is towards the flexible side of the building, considering the in-plan position of the vertical ideal principal axis $III_{sec}(CR_{sec})$ as the origin. The relative in-plan positions of the vertical $III_{sec}(CR_{sec})$ axis and CM across each floor-plan designate the edge sides of the multi-story building as flexible or stiff. All lateral floor forces are applied relative to the "Capable Near Collapse Principal System, $III_{sec}(CR_{sec})$, $I_{sec}$, $II_{sec}$" of the multi-story RC building. In Figure 2, we can see the floor lateral loading vectors $\mathbf{P}_{flex,Isec}$, $\mathbf{P}_{flex,IIsec}$ and $\mathbf{P}_{stiff,Isec}$, $\mathbf{P}_{stiff,IIsec}$ applied on each *i*-floor at the positions 3, 1 and 4, 2 in order to safely estimate the ductility demands at the flexible and stiff sides of the multi-story RC building, respectively, along the $I_{sec}$ or $II_{sec}$ axis.

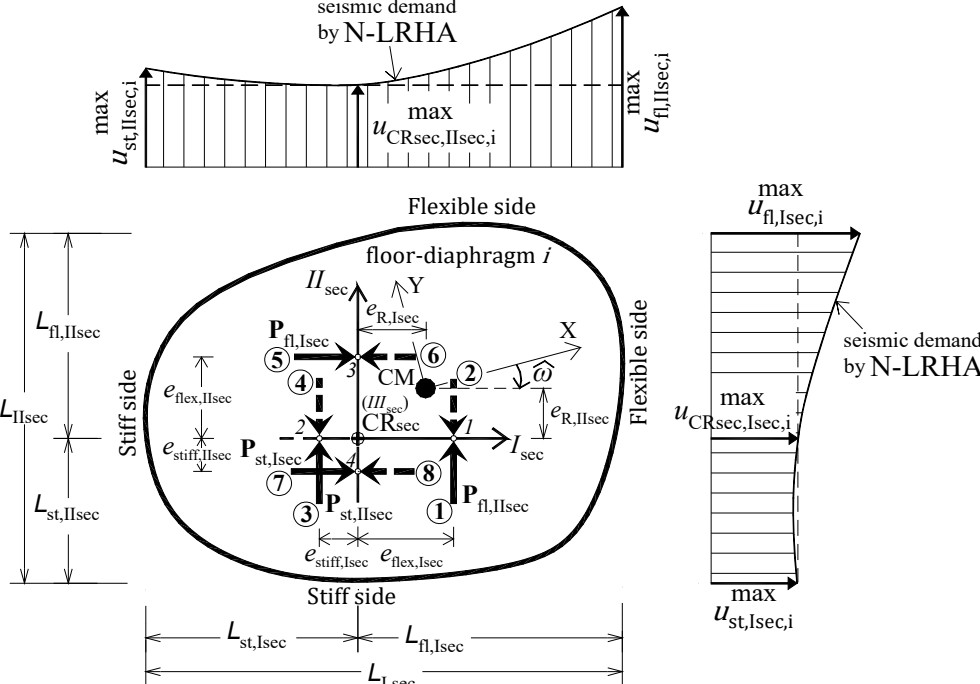

**Figure 2.** Application of the lateral static loads on each *i*-floor at two different positions determined by the inelastic dynamic eccentricities $e_{flex}$ and $e_{stiff}$ along each horizontal principal direction. Positions 3, 1 and 4, 2 are used, respectively, to safely estimate the ductility demands at the flexible and stiff sides of the multi-story building along the $I_{sec}$ and $II_{sec}$ axes, when the accidental eccentricity is not considered in analysis.

The appropriate inelastic dynamic eccentricities have been found from a large parametric analysis on single-story RC buildings [4] and are given by graphs (Figure 3) and equations (Equations (10)–(13)), using as parameters the inelastic static eccentricity $e_R$, the radius of gyration $r_m$ of the floor, and the category of inelastic torsional sensitivity of the single-story building. These dynamic eccentricities are also verified in an extended

parametric analysis of multi-story RC buildings [9], where it is shown that, together with the use of appropriate load patterns, the seismic ductility demands of multi-story RC buildings can be safely predicted. In this parametric analysis, various structural types of ductile multi-story RC buildings with various numbers of floors were examined by performing N-LRHA. The parameters investigated were the magnitude of the inelastic static eccentricity and the torsional sensitivity category of multi-story RC buildings, as defined in Section 2.2.

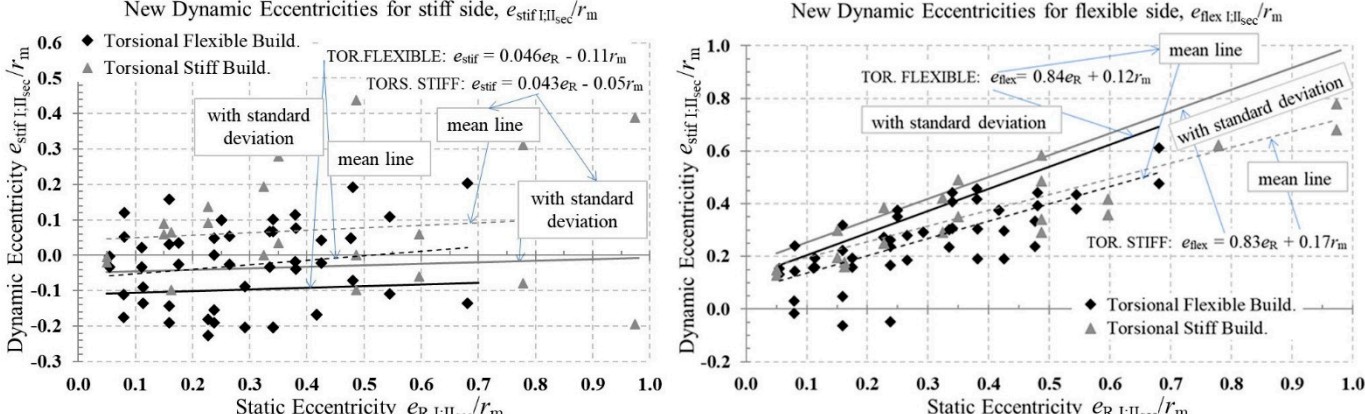

**Figure 3.** Normalized inelastic dynamic eccentricities, left: $e_{\text{stif I;IIsec}}/r_{\text{m}}$ for the stiff side, right: $e_{\text{flex I;IIsec}}/r_{\text{m}}$ for the flexible side.

For torsionally sensitive buildings, i.e., when $r_{\text{I,sec}}$ or $r_{\text{II,sec}} \leq 1.10\, r_{\text{m}}$ applies:

$$e_{\text{stiff},i} = 0.046 \cdot e_{\text{R},i} - 0.11 \cdot r_{\text{m}} \tag{10}$$

$$e_{\text{flex},i} = 0.84 \cdot e_{\text{R},i} + 0.12 \cdot r_{\text{m}} \tag{11}$$

For torsionally non-sensitive buildings, i.e., when $r_{\text{I,sec}}$ and $r_{\text{II,sec}} > 1.10\, r_{\text{m}}$ applies:

$$e_{\text{stiff},i} = 0.043 \cdot e_{\text{R},i} - 0.05 \cdot r_{\text{m}} \tag{12}$$

$$e_{\text{flex},i} = 0.83 \cdot e_{\text{R},i} + 0.17 \cdot r_{\text{m}} \tag{13}$$

where the subscript *i* refers to the direction of $I_{\text{sec}}$ or $II_{\text{sec}}$ axis, $e_{\text{R},i}$ is the inelastic static eccentricity, and $r_{\text{m}}$ is the radius of gyration of the floor-diaphragm closest to the $0.8H_{\text{tot}}$ level, as defined in Section 2.2. The torsional sensitivity of the multi-story building is verified by Equation (9).

### 2.4. Handling of Accidental Eccentricities. Definition of Inelastic Design Eccentricities

If accidental eccentricities are also considered in analysis, then they are combined with the inelastic dynamic eccentricities in the most unfavorable way, to apply the lateral static forces in each floor-diaphragm more eccentric relative to CM (Figure 4). Again, the "Capable Near Collapse Principal System, $III_{\text{sec}}(\text{CR}_{\text{sec}})$, $I_{\text{sec}}$, $II_{\text{sec}}$" of the multi-story RC building is used as the reference system. The resulted eccentricities are called the inelastic design eccentricities and are given by Equations (14)–(17):

$$e_1 = e_{\text{flex},I\,\text{sec}} + e_{\text{a},I\,\text{sec}} \tag{14}$$

$$e_2 = e_{\text{stiff},I\,\text{sec}} - e_{\text{a},I\,\text{sec}} \tag{15}$$

$$e_3 = e_{\text{flex},II\,\text{sec}} + e_{\text{a},II\,\text{sec}} \tag{16}$$

$$e_4 = e_{\text{stiff},II\,\text{sec}} - e_{\text{a},II\,\text{sec}} \tag{17}$$

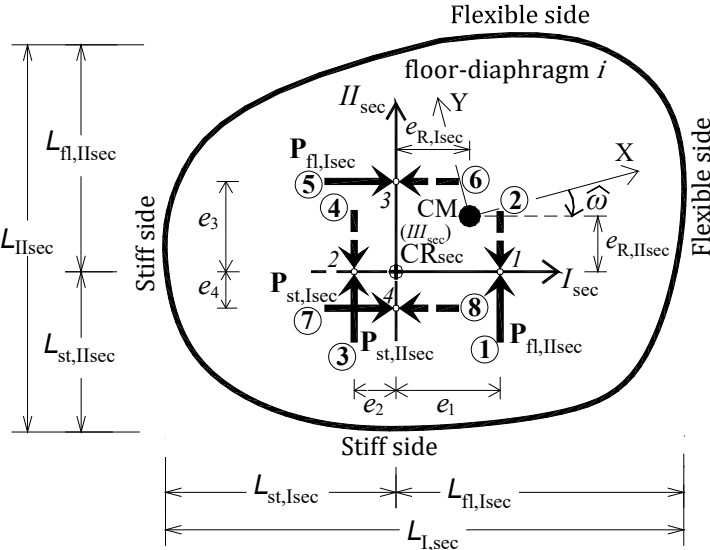

**Figure 4.** Application of the lateral static loads on each *i*-floor at two different positions along each horizontal principal direction $I_{sec}$ and $II_{sec}$ (1, 2 and 3, 4) determined by the inelastic design eccentricities $e_1$, $e_2$ and $e_3$, $e_4$, respectively, when the accidental eccentricity is also considered in analysis.

In Equations (14)–(17), $e_{stiff,I\,sec}$, $e_{flex,I\,sec}$ and $e_{stiff,II\,sec}$, $e_{flex,II\,sec}$ are the inelastic dynamic eccentricities (Equations (10) and (11) or Equations (12) and (13)) and $e_{a,I\,sec}$, $e_{a,II\,sec}$ are the accidental eccentricities along the principal directions $I_{sec}$ and $II_{sec}$, respectively. The latter are calculated by the equations $e_{a,Isec} = \pm(0.05 \sim 0.10) \cdot L_{Isec}$ and $e_{a,IIsec} = \pm(0.05 \sim 0.10) \cdot L_{IIsec}$ according to EN1998-1, where $L_{Isec}$, $L_{IIsec}$ are the maximum dimensions of the floors normal to the loading direction. In Figure 4, we can see the floor lateral loading vectors $\mathbf{P}_{flex,Isec}$, $\mathbf{P}_{flex,IIsec}$ and $\mathbf{P}_{stiff,Isec}$, $\mathbf{P}_{stiff,IIsec}$ applied on each *i*-floor at the positions 3, 1 and 4, 2 using the design eccentricities $e_3$, $e_1$ and $e_4$, $e_2$, respectively, to safely estimate the ductility demands at the flexible and stiff sides of the multi-story RC building, along the $I_{sec}$ or $II_{sec}$ axis.

*2.5. Consideration of the Higher-Mode Effects*

The lateral static forces are applied on each floor, along each vertical loading principal plane defined by the inelastic dynamic or design eccentricities, using two patterns, as shown in Figure 5 [9]. The first one is proportional to the fundamental translational mode-shapes along the principal directions $I_{sec}$ and $II_{sec}$ corresponding to a unit base shear (Equation (18)). The second one is again proportional to the fundamental translational modes but now is calculated with a reduced base shear while the remainder magnitude of the base shear is applied at the top of the building as an additional force. This is behind the rationale of some seismic codes [36–38]. The main purpose of the second loading pattern is to rationally consider the higher-mode effects. The additional top force is defined with a magnitude equal to 20% of the unit base shear and the remainder 80% of the unit base shear is used to calculate the distributed along the height lateral floor forces according to the uncoupled mode-shapes (Equation (19)).

$$F_i = m_i \varphi_i \text{ for } i = 1 \text{ to } n \,, \quad \overline{F}_i = F_i / \sum_i^n F_i = m_i \varphi_i / \sum_i^n m_i \varphi_i \,, \quad \sum_i^n \overline{F}_i = V_b = 1 \qquad (18)$$

$$F_i = \begin{cases} 0.80 m_i \varphi_i \,, & i = 1 \text{ to } n-1 \\ 0.80 m_i \varphi_i + F_H \,, & i = n \end{cases} \quad, \overline{F}_i = F_i / \sum_i^n F_i \,, \quad \sum_i^n \overline{F}_i = V_b = 1 \qquad (19)$$

where the additional top force is equal to $F_H = 0.20 \sum_{j=1}^{n} m_j \varphi_j$ (giving a normalized additional top force $\overline{F}_H = 0.20$), $m_i$ is the concentrated mass of the $i$-floor, $\varphi_i$ is the modal component of the $i$-floor under the fundamental translational mode along the $I_{sec}$ or $II_{sec}$ axis, and $\overline{F}_i$ is the normalized value of the lateral force of the $i$-floor acting along the direction of $I_{sec}$ or $II_{sec}$ axis. The normalization of the lateral floor forces provides a unit base shear for the building. Equations (18) and (19) are valid for various normalization forms of the fundamental mode-shapes, i.e., even for normalization that provides $\varphi_n = 1$.

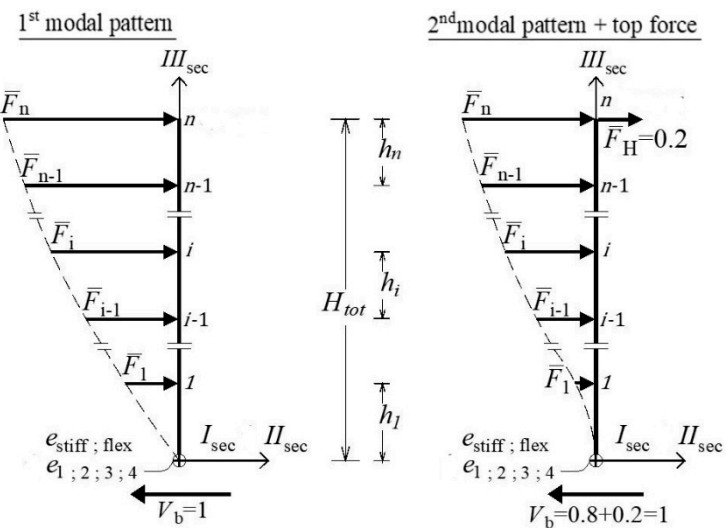

**Figure 5.** Vertical loading principal planes defined by the inelastic dynamic or design eccentricities along the $I_{sec}$ or $II_{sec}$ axes. Two patterns of floor lateral static forces are used in the proposed pushover procedure: an uncoupled translational modal one for each direction $I_{sec}$ and $II_{sec}$ and a modal one with a reduced base shear but with an additional top force.

### 2.6. Target Displacement of the Proposed Pushover Analysis at NC and Capacity Curves

The application of the floor lateral static forces according to Section 2.3 or Sections 2.4 and 2.5, with two signs ($\pm$) of action as shown in Figures 2 and 4, leads to eight (8) separate pushover analyses to be performed per load pattern (numbered as 1–8 in circle). The target displacement at the building top to be reached in each one of the sixteen separate pushover analyses (for the two load patterns) at the Near Collapse (NC) state is computed from Annex B of EN 1998-1 or by N-LRHA. For an alternative estimation of the target displacement, Table 1 shows the proposed mean values of the seismic target angular deformation $\gamma_{t,top}$ of the building (equal to the lateral displacement of the top floor divided by the total building height) derived from the extended parametric analysis of ductile multi-story RC buildings by performing N-LRHA [9]. The monitoring point coincides with the intersection of the vertical ideal principal axis $III_{sec}$ with the diaphragm of the top floor. Since the in-plan positions of the applied lateral static forces at the building top, i.e., those determined by the inelastic dynamic or design eccentricities, are different from the in-plan position of $III_{sec}$ axis, these proposed values should be modified. However, as observed from the N-LRHA displacement profile envelopes at the NC state, the difference between the seismic displacement on $III_{sec}$ axis and those corresponding to the loading in-plan positions is small, especially when the static eccentricity is low (e.g., up to $0.15L_{I;IIsec}$). Hence, the proposed values of Table 1 are approximately valid for all ductile multi-story buildings with moderate static eccentricity. For a more accurate determination of the NC target displacement at the various loading positions, the one that corresponds to the flexible sides ($e_{flex,Isec;IIsec}$ or $e_1$, $e_3$) can be found from the corresponding pushover analyses when the Near Collapse state of the building is shown while the other that corresponds to the stiff sides ($e_{stiff,Isec;IIsec}$ or $e_2$, $e_4$) can be taken from Table 1. It should be noted that the capacity

curves of the multi-story buildings along the horizontal ideal principal axes ($I_{sec}$ or $II_{sec}$) are determined by the base shear and the lateral displacement of the monitoring point along the examined direction. The latter is the point of application of the lateral forces on the top floor.

**Table 1.** Seismic target angular deformation $\gamma_{t,top}$ of the building at the NC state (mean values), on the top of the vertical axis $III_{sec}$ and along the horizontal axes $I_{sec}$ and $II_{sec}$. Values for dual systems (frames and walls) or coupled (via beams) wall systems, as well as for different number of stories, can be found by linear interpolation.

| Number of Stories | 1 | 2 | 3 | 4 | 5 |
|---|---|---|---|---|---|
| Pure frame buildings without walls | 0.0300 | 0.0295 | 0.0235 | 0.0205 | 0.0195 |
| Pure wall buildings without frames | 0.0280 | 0.0290 | 0.0260 | 0.0240 | 0.0230 |

Note: $\gamma_{t,top}$ mean values are proposed for all asymmetric buildings regardless of torsional sensitivity category.

*2.7. Evaluation of the Seismic Demand at the NC State Due to the Spatial Seismic Action*

The concurrent action of the two horizontal components of the seismic action is taken into account by sixteen (16) SRSS combinations of the effects of the eight (8) separate pushover analyses per load pattern (numbered as 1–8 in circle, in Figures 2 and 4), as in EN 1998–1. In other words, the response effects resulted from the separate pushover analyses 5, 6 and 7, 8 along the $I_{sec}$ axis are combined with the corresponding ones resulted from the separate pushover analyses 1, 2 and 3, 4 along the $II_{sec}$ axis with the following 16 SRSS (symbol $\oplus$) combinations:

$$(5) \oplus (1), \ (5) \oplus (2), \ (6) \oplus (1), \ (6) \oplus (2), \ (7) \oplus (1), \ (7) \oplus (2), \ (8) \oplus (1), \ (8) \oplus (2)$$

and

$$(5) \oplus (3), \ (5) \oplus (4), \ (6) \oplus (3), \ (6) \oplus (4), \ (7) \oplus (3), \ (7) \oplus (4), \ (8) \oplus (3), \ (8) \oplus (4).$$

The envelope of a total of thirty-two (32) SRSS combinations of the response effects (displacement, deformation, stress below yield point) for the two load patterns, can be considered as a rational estimate of the seismic demands due to the spatial seismic action.

*2.8. Significant Damage (SD) and Damage Limitation (DL) Performance Levels: Verification Using the Proposed Procedure*

For the seismic assessment of multi-story buildings at the performance levels of Significant Damage (SD) and Damage Limitation (DL) according to Eurocode 1998-3 (better known as the "Life Safety State" and "Operational State", respectively) [9], the "inelastic dynamic eccentricities" pushover is performed again with the following differentiations:

(a) For the verification of the building at the DL state, it is suggested to provide each structural member of the nonlinear model with the effective bending stiffness equal to $0.25EI_g \leq EI_{eff,DL} = 2EI_{sec} \leq 0.5EI_g$, where $I_g$ is the is the moment of inertia of the geometric section. The target displacement, at the point of application of the lateral loading on the top floor of the building, is determined from response spectrum analysis or N-LRHA (or even LRHA) or with the use of the informational Annex B of EN 1998-1 for the DL earthquake or from the reduced by 75% values determined from the alternative estimation of the NC target displacement in Section 2.6.

(b) For the verification of the building at the SD state, it is suggested to provide each structural member of the nonlinear model with the effective bending stiffness equal to the average of the corresponding values used for the DL and NC states, i.e., the average of $EI_{eff,DL}$ and $EI_{sec}$. The target displacement, at the point of application of the lateral loading on the top floor of the building, is determined from N-LRHA or with the use of informational Annex B of EN 1998-1 for the SD earthquake or from the reduced by 30% values determined from the alternative estimation of the NC target displacement in Section 2.6.

All the above-mentioned suggestions derive from the extended parametric analysis of ductile multi-story RC buildings [9], which is carried out in the context of the doctoral dissertation of the first author.

## 3. Numerical Example of a Six-Story Building

The applications steps of the proposed pushover procedure are presented in detail herein by the seismic assessment of a six-story, double-asymmetric, torsionally sensitive, RC building.

### 3.1. Building Description

The six-story RC building shown in Figure 6 is a dual system consisting of two walls coupled with frames. The building layout has a trapezoidal shape, having two perimetric frames along the *x*-axis with the longer one coupled with the weakest wall, two perimetric frames along the inclined sides of the trapezoid with one of them coupled with the strongest wall and an interior frame along the *y*-axis starting from the boundary element (barbell) of the weakest wall (Figure 6). A rigid diaphragm of thickness 0.17 m extends out of the building layout forming a 2 m perimetric cantilever. The mass and the mass moment of inertia of each floor are considered equal to 250 tn and 12000 tn·m$^2$, respectively. The Center of Mass (CM) of each floor coincides with its geometrical center (CG) and all CM lie on the same vertical axis. The concrete grade is C30/37 of mean compressive strength equal to 38 MPa while the reinforcement steel grade is B500c of mean strength 550 MPa. All columns have a square section of dimension 0.55 m from the base of the building to the third floor and 0.5 m from the third floor to the top. The beams are considered having T section with flange 1.5/0.17 m and web 0.3/0.6 m for stories 1 to 4 and 0.3/0.55 m for stories 5–6. The beams have a similar section in all frames except in one of the inclined sides which has T beams with a wider web of dimension 0.40 m. The two walls are of orthogonal shape of dimensions 1.2/0.3 and 1.5/0.3 m with the smaller one (along the *x*-axis) having a boundary barbell 0.45/0.45 m. The total height of the building is 21.5 m, the height of the first floor is 4 m, while the height of the other five stories is 3.5 m.

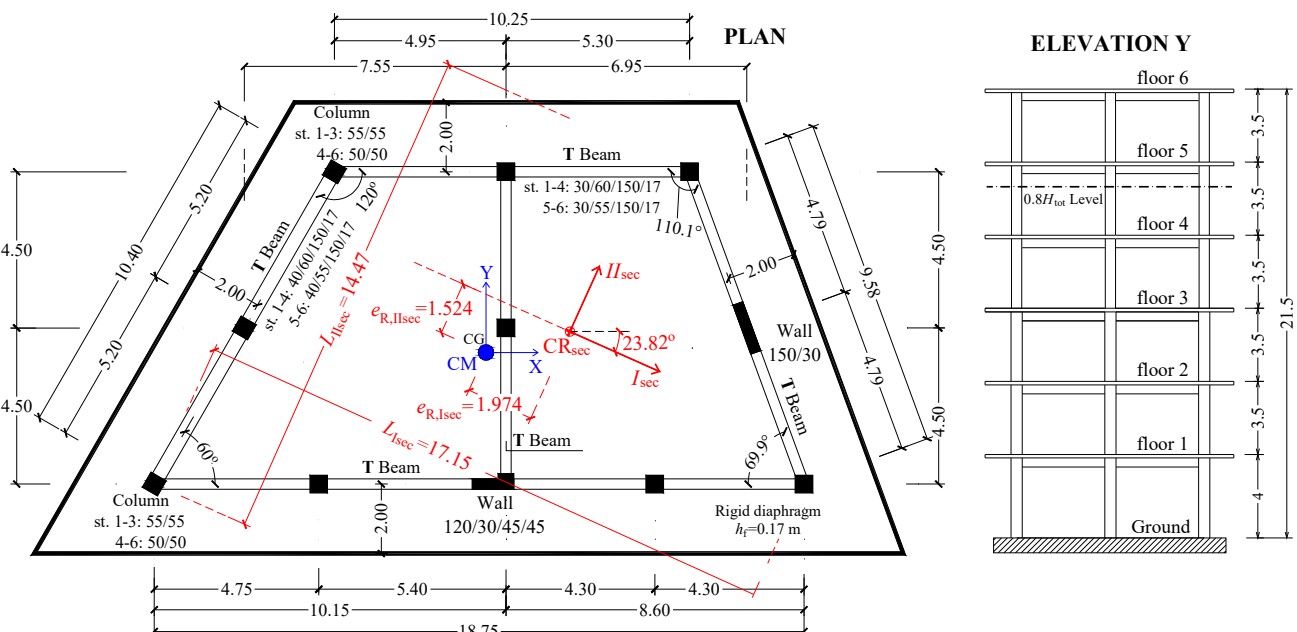

**Figure 6.** Plan and elevation view of the six-story RC building. "Capable Near Collapse Principal System, $III_{sec}(CR_{sec})$, $I_{sec}$, $II_{sec}$" defined in the nonlinear model ($EI_{sec}$) of the building.

### 3.2. Design of the Six-Story Building

The six-story RC building is designed for ductility class high (DCH) according to the provisions of Eurocodes EN1992-1 and EN1998-1 as an ordinary building (importance factor $\gamma_1 = 1$), with effective peak ground acceleration $\alpha_g = 0.16$ g, soil category D and total behavior factor $q = 4$. The linear model of the building was analyzed by performing modal response spectrum analysis. In the design process, all the structural elements of the linear model of the building have been provided with their effective flexural and shear stiffness that is equal to one-half of their respective uncracked (geometric) stiffness. The building is classified into the structural type of dual buildings, equivalent to wall buildings, along both the x, y-axes, according to EN1998-1. Additionally, the building is characterized as torsionally non-sensitive since both the torsional radii ratios, $r_{II,des}/r_m = 1.03$ and $r_{I,des}/r_m = 1.10$, are greater than 1. The translational uncoupled periods of the building are 1.06 sec along the $II_{des}$ axis and 0.99 sec along the $I_{des}$ axis. The horizontal ideal principal axes $I_{des}$ and $II_{des}$ of the building are rotated by $-17.5°$ relative to the x, y-axes and the double static eccentricity $e_{I;IIdes}$ (distance between $CR_{des}$ and CM in the floor-diaphragm closest to the $0.8H_{tot}$ level from the base) is about equal to $0.10L_{I;IIdes}$ along both the horizontal ideal principal axes. The designed building has appropriate longitudinal/confinement steel reinforcement details that provide an overall high ductile behavior, spreading the ductility demands to the end-sections of all beams and to the base end-sections of all columns/walls (beam-sway mechanism). The numbering of the structural members in the mathematical model of the building, the section properties and the reinforcement details are presented in Figure A1 and in Tables A1–A3 of Appendix A.

### 3.3. Non-Linear Model

The non-linear model is created according to Section 2.1, by providing the secant stiffness $EI_{sec}$ at yield to all structural elements. The section analysis has been performed by the FEM program SAP2000 (module Section Designer) [39]. The mean values of the ratio of the secant stiffness at yield (Equation (1)) to the corresponding geometric stiffness of the structural elements ($EI_{sec}/EI_g$) in each floor are reported in Table 2, separately for the columns, walls and beams, which are modeled as 3D Frame elements (6 DoFs per joint). Point $M_3$ and $P$-$M_2$-$M_3$ hinges are inserted at the end-sections of beams and columns/walls, respectively, with constitutive laws according to Mander et al. [40] for the unconfined/confined concrete and according to Park [41] for the steel reinforcement. The plastic hinge length $L_{pl}$ (Equation (4)) divided by the $\gamma_{el}$ factor is provided to the analysis software to convert the $M$-$\varphi$ curves to $M$-$\theta$ ones. The "Capable Near Collapse Principal System, $III_{sec}(CR_{sec}), I_{sec}, II_{sec}$" of the six-story RC building is determined according to Section 2.2 at the floor-diaphragm closest to the level $0.8H_{tot} = 17.20$ m from the building base, i.e., at the fifth floor with height equal to 18 m measured from the building base. The in-plan position of the "inelastic" center of stiffness $CR_{sec}$ and the orientation of the horizontal ideal "inelastic" principal axes $I_{sec}$, $II_{sec}$ of the building are shown in Figure 6. The latter are turned relative to x, y axes by 23.82° clockwise. The inelastic static eccentricities are equal to $e_{R,Isec} = 1.974$ m and $e_{R,IIsec} = 1.524$ m and their normalized values are equal to $e_{R,Isec}/L_{Isec} = 0.12$ and $e_{R,IIsec}/L_{IIsec} = 0.11$, where $L_{Isec}$ and $L_{IIsec}$ are the maximum plan dimensions along the axes $I_{sec}$ and $II_{sec}$, respectively. The (mean) normalized "inelastic" torsional radii are equal to $r_{I,sec}/r_m = 1.104$ and $r_{I,sec}/r_m = 1.033$, where $r_m = 6.928$ m is the radius of gyration of the floor-mass at the fifth floor.

**Table 2.** Mean values of the ratio $EI_{sec}/EI_g$ of the secant stiffness at yield to the geometric stiffness in each story, separately for the columns, walls, and beams. The local axes of structural elements are denoted by 2 and 3, where 3 is normal to their strong direction.

| | **Mean Values of 9 Columns** | | **Mean Values of 2 Walls** | | **Mean Values of 12 Beams** |
|---|---|---|---|---|---|
| **Story** | $EI_{3,sec}/EI_{3,g}$ | $EI_{2,sec}/EI_{2,g}$ | $EI_{3,sec}/EI_{3,g}$ | $EI_{2,sec}/EI_{2,g}$ | $EI_{3,sec}/EI_{3,g}$ |
| 1 | 0.15 | 0.17 | 0.31 | 0.27 | 0.128 |
| 2 | 0.13 | 0.15 | 0.29 | 0.25 | 0.129 |
| 3 | 0.14 | 0.14 | 0.20 | 0.20 | 0.117 |
| 4 | 0.15 | 0.15 | 0.16 | 0.19 | 0.104 |
| 5 | 0.13 | 0.13 | 0.12 | 0.17 | 0.088 |
| 6 | 0.10 | 0.10 | 0.08 | 0.14 | 0.082 |

The building is classified as torsionally sensitive according to Equation (9) (for the torsional verification in the non-linear area), because the smaller of the two torsional radii ratios (1.033) is lower than 1.10. This is the first case (case 1) of torsional sensitivity that will be examined. The first three uncoupled modes of the "case 1" nonlinear model have periods equal to 1.87, 1.75 and 1.70 s, where the first and second ones are translational along the $II_{sec}$ and $I_{sec}$ axis, respectively, while the third one is torsional around $z$-axis. The second case of torsional sensitivity (case 2) that will be examined comes from an artificial increase of the mass moment of inertia of each floor to the value of 20,000 tn·m², which is by 67% higher from the nominal value (12,000 tn·m²). Therefore, this case refers to a more torsionally sensitive building where the radius of gyration of the floor-mass becomes equal to $r_m = 8.944$ m and the smaller of the two torsional radii ratios, $r_{I,sec}/r_m = 0.855$ and $r_{I,sec}/r_m = 0.80$, is well below the limit value 1.10 of Equation (9). The first three uncoupled modes of the "case 2" nonlinear model have periods equal to 2.19, 1.87, and 1.75 s, where the first one is torsional around $z$-axis while the second and third ones are translational along the $II_{sec}$ and $I_{sec}$ axis, respectively. Accidental eccentricity will not be considered in this numerical example.

### 3.4. Calculation of Inelastic Dynamic Eccentricities

Using the abovementioned data, the inelastic dynamic eccentricities are calculated by Equations (10) and (11) for torsionally sensitive buildings, according to Section 2.3:

Case 1 ($r_m = 6.928$ m):

$$e_{stif,Isec} = 0.046 \cdot e_{R,Isec} - 0.11 \cdot r_m = 0.046 \cdot 1.974 - 0.11 \cdot 6.928 = -0.671 \text{ m}$$
$$e_{stif,IIsec} = 0.046 \cdot e_{R,IIsec} - 0.11 \cdot r_m = 0.046 \cdot 1.524 - 0.11 \cdot 6.928 = -0.692 \text{ m}$$
$$e_{flex,Isec} = 0.84 \cdot e_{R,Isec} + 0.12 \cdot r_m = 0.84 \cdot 1.974 + 0.12 \cdot 6.928 = 2.490 \text{ m}$$
$$e_{flex,II sec} = 0.84 \cdot e_{R,IIsec} + 0.12 \cdot r_m = 0.84 \cdot 1.524 + 0.12 \cdot 6.928 = 2.111 \text{ m}$$

Case 2 ($r_m = 8.944$ m):

$$e_{stif,Isec} = 0.046 \cdot e_{R,Isec} - 0.11 \cdot r_m = 0.046 \cdot 1.974 - 0.11 \cdot 8.944 = -0.893 \text{ m}$$
$$e_{stif,IIsec} = 0.046 \cdot e_{R,IIsec} - 0.11 \cdot r_m = 0.046 \cdot 1.524 - 0.11 \cdot 8.944 = -0.914 \text{ m}$$
$$e_{flex,Isec} = 0.84 \cdot e_{R,Isec} + 0.12 \cdot r_m = 0.84 \cdot 1.974 + 0.12 \cdot 8.944 = 2.732 \text{ m}$$
$$e_{flex,II sec} = 0.84 \cdot e_{R,IIsec} + 0.12 \cdot r_m = 0.84 \cdot 1.524 + 0.12 \cdot 8.944 = 2.353 \text{ m}$$

The "Capable Near Collapse Center of Stiffness" $CR_{sec}$ is the origin for the measurement of inelastic dynamic eccentricities inside each floor-diaphragm along the "Capable Near Collapse Principal Axes" $I_{sec}$ and $II_{sec}$, with positive direction towards CM. Hence, the negative sign of the inelastic dynamic eccentricities for the stiff sides indicates that the lateral static forces should apply on each $i$-floor towards the stiff side of the building, relative to $CR_{sec}$.

### 3.5. Application of the Floor Lateral Forces in Plan and in Elevation

Using the inelastic dynamic eccentricities calculated in the previous section, the lateral static forces are applied at the in-plan positions 3, 1 and 4, 2 inside each floor (as in Figure 2) to safely predict the floor displacements and the floor angular deformations at the flexible and stiff sides of the building, respectively (Section 2.3). The positions of the lateral static forces in all floors are illustrated in Figure 7, where the eight (8) separate ($\pm$) pushover analyses per pattern that should be performed are numbered in a circle as 1 to 8. Accidental eccentricities are not considered in this numerical example. According to Section 2.5, two patterns of floor lateral forces are applied in elevation, along the vertical principal planes defined by the inelastic dynamic eccentricities. Both patterns shown in Figure 8 are proportional to the fundamental uncoupled translational modes along the principal directions $I_{\text{sec}}$ and $II_{sec}$, but the second one is calculated for 80% of the base shear and the remainder 20% is applied as an additional top force in order to consider the higher-mode effects (Equations (18) and (19)). It is noted that the translational mode-shapes along the principal directions $I_{\text{sec}}$ and $II_{sec}$ are practically identical.

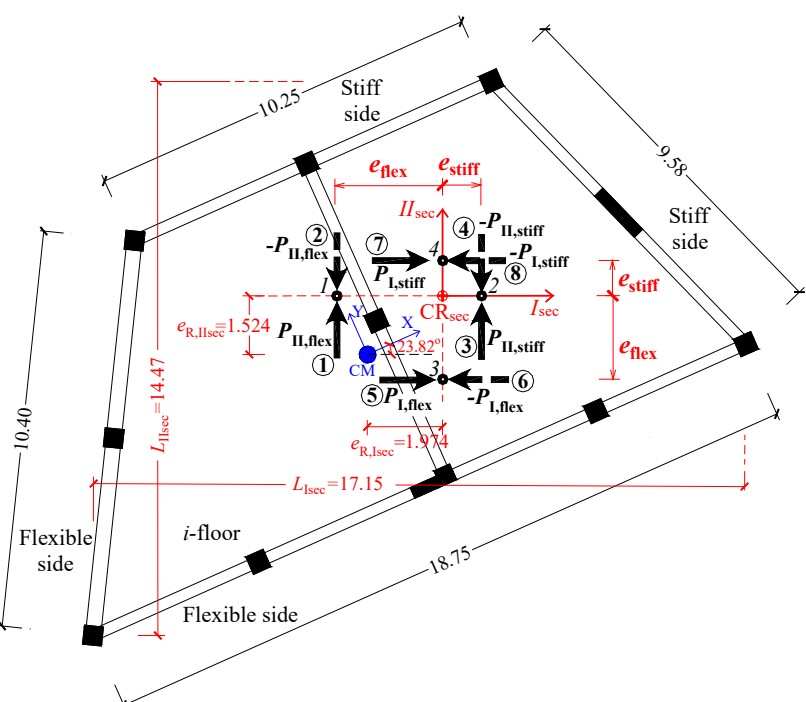

**Figure 7.** Eight (8) pushover analyses per pattern. The floor lateral static forces are applied at the positions determined by the inelastic dynamic eccentricities $e_{\text{flex}}$ and $e_{\text{stiff}}$.

### 3.6. Target Displacement of the Eight Separate Pushover Analysis (Per Pattern) by N-LRHA

To perform the proposed pushover procedure, the target displacement at the positions of the applied lateral static loads inside the top floor-diaphragm must be known (Section 2.6). In the framework of this example, the target displacement is determined by performing N-LRHA using three pairs of horizontal accelerograms (Figure 9a). These pairs were defined by using five "unit" artificial uncorrelated accelerograms created by Seismoartif [42]. All accelerograms have similar characteristics with the Hellenic tectonic faults as well as the main specifications of earthquakes recorded in Greece [43]. Their elastic acceleration response spectra are approximately equal to the design acceleration spectrum of EN 1998-1 for soil class D (Figure 9b).

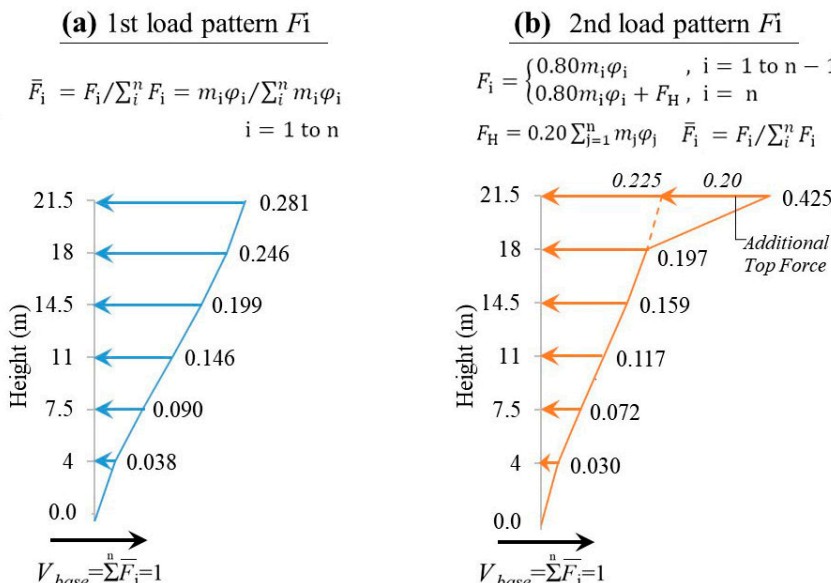

**Figure 8.** Two different in elevation loading patterns: (**a**) proportional to the fundamental uncoupled translational modes along $I_{sec}$ and $II_{sec}$ axis for a base shear equal to 1 kN, (**b**) similar to the first but for 80% of the unit base shear and the remainder 20% is applied as an additional lateral force at the building top.

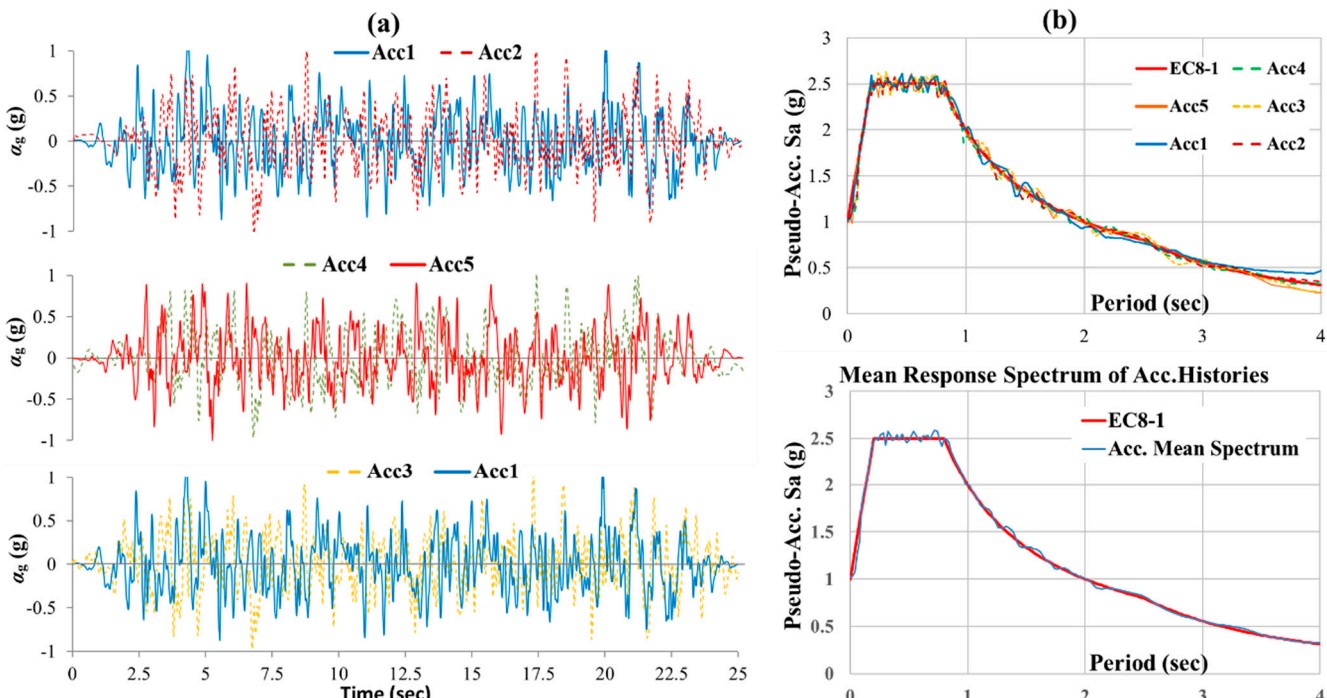

**Figure 9.** (**a**) Three pairs of unit-normalized artificial accelerograms ($a_g \cdot S = 1.00 \cdot g$, $t_d = 25$ s, strong motion duration 19 s), (**b**) Elastic acceleration spectra of the 5 accelerograms and their mean acceleration spectrum relative to the design spectrum of EN 1998-1 (damping 0.05, $a_g \cdot S = 1 \cdot g$ and soil D).

The two horizontal accelerograms of each pair are scaled to a PGA value equal to 0.39 g that causes the Near Collapse (NC) state of the building. Each pair is rotated about the vertical axis successively per 22.5° in order to find the most unfavorable loading state [44]. Additionally, the two seismic components of each pair were examined with all combinations of signs (i.e., $++$, $+-$, $-+$ and $--$). Finally, a total of 192 N-LRHA

are performed, and the envelope of the displacement demands along the axes $I_{sec}$ and $II_{sec}$ is considered as the "seismic target-displacement" for each monitoring point in the floor-plans.

Instead, if the seismic target displacement at the NC state is calculated by the proposed seismic target angular deformation of the building in Table 1, a value of $u_t = \gamma_{t,top}H_{tot} = 0.0243\cdot21.50 = 0.52$ m at the top of $III_{sec}$ axis (CR$_{sec}$) along both the $I_{sec}$ and $II_{sec}$ axes is obtained, where $\gamma_{t,top} = 0.0243$ rad is taken from Table 1 by linear interpolation of the proposed values for five and ten-story pure frame and wall buildings. This seismic target displacement differs very little from the corresponding values of about 0.53 m (case 1) or 0.54 m (case 2) obtained by N-LRHA. As we can see in Table 3, the target displacement at the top of $III_{sec}$ axis by Table 1 is also close enough to the corresponding displacements resulted by the N-LRHA at the in-plan positions of the lateral loading (points 3, 1 and 4, 2, respectively, in Figure 7). Additionally, Table 3 shows the seismic target displacement at the top of $III_{sec}$ axis (CR$_{sec}$) along the $I_{sec}$ and $II_{sec}$ axes calculated by the informational Annex B of EN 1998-1. It is noted that in order to apply the latter, the corresponding pushover curves should be first bi-linearized.

**Table 3.** Comparison of target displacement at the top of the building.

| Target Displacement | By Table 1 | N-LRHA (0.39 g) | | | | | | Inf. Annex B EN 1998-1 |
|---|---|---|---|---|---|---|---|---|
| | | Case 1 | Case 2 | Case 1 | | Case 2 | | |
| (m) | $III_{sec}$ Axis Top | $III_{sec}$ Axis Top | $III_{sec}$ Axis Top | Point 3/1 | Point 4/2 | Point 3/1 | Point 4/2 | $III_{sec}$ Axis Top |
| $u_{t,Isec}$ | 0.52 | 0.53 | 0.54 | 0.54 | 0.53 | 0.54 | 0.57 | 0.462 |
| $u_{t,IIsec}$ | | 0.53 | 0.55 | 0.56 | 0.53 | 0.55 | 0.55 | 0.50 |

### 3.7. Verification Procedure

The envelope of the thirty-two (32) combined effects by the SRSS rule of the sixteen (16) pushover analyses using inelastic dynamic eccentricities (eight (8) per pattern) is considered as representative of the spatial seismic action at the NC state (Section 2.7). The proposed pushover procedure will be verified relative to the results of N-LRHA. The comparison will be performed in terms of floor displacements, floor angular deformations, plastic chord rotations, and story shears. The verification of the building at the NC state is performed according to EN 1998-1, at the attainment of D/C (Demand to Capacity) ratios equal to one in terms of chord rotations of the structural members at ultimate state.

Particular attention should be given to the floor angular deformations, which determine the seismic ductility demands and therefore are responsible for the damage potential. In this respect, the inelastic floor angular deformations along four vertical bending planes per principal direction $I_{sec}$ or $II_{sec}$ resulted from the EN 1998-1 pushover (N2) [45], the extended N2 pushover [19] and the "corrective eccentricities" pushover [26] will also be presented.

The "corrective eccentricities" pushover is conceptually similar to the proposed pushover procedure but uses a completely different methodology. According to this method, the floor lateral static loads are applied using a corrective eccentricity (relative to CM), plus any accidental one, only for the estimation of the ductility demands at the stiff sides. For the estimation of the ductility demands at the flexible sides, the floor lateral static loads are applied at the position of CM in each floor, plus any accidental eccentricity if it is considered in analysis, as in EN 1998-1. It is noted that the "corrective eccentricities" method is performed considering the same patterns of floor lateral static loads in elevation used in the pushover analysis according to the proposed procedure and again the envelope of the thirty-two (32) SRSS combinations of the effects of the sixteen (16) "corrective eccentricities" pushover analyses is considered as the seismic demand.

Additionally, the N2 pushover procedure is performed according to EN 1998-1, by applying the (first) modal pattern of lateral static forces at CM in each floor since the accidental eccentricity is not considered in this example. Then, the envelope of the four

SRSS combinations of the effects of the four separate ($\pm$) pushover analyses (two along each principal direction) is considered as the seismic demand. It is noted that the uniform pattern proposed by EN 1998-1 is not used in the pushover analysis because it provides very conservative estimates on the response of lower floors.

Further, the extended N2 procedure of pushover analysis is performed. According to this procedure, the response effects (displacements, deformations, chord rotations, stress) resulted by the four separate ($\pm$) pushover analyses along each principal direction of the N2 [45] procedure are corrected using corrections factors, in plan and in elevation, determined by the results of a 3D modal response spectrum analysis.

Additionally, the floor angular deformation results by a recently developed pushover procedure with seismic floor enforced displacements [9] are also presented. According to this procedure, the seismic demand is estimated by the envelope of sixteen (16) pushovers with appropriate simultaneous combinations of two floor enforced translational displacements along the horizontal $I_{sec}$ and $II_{sec}$ axes and one floor enforced rotation about the vertical $III_{sec}$ axis, acting inside each floor-diaphragm at the position of the vertical ideal principal axis $III_{sec}$ (Figure 10).

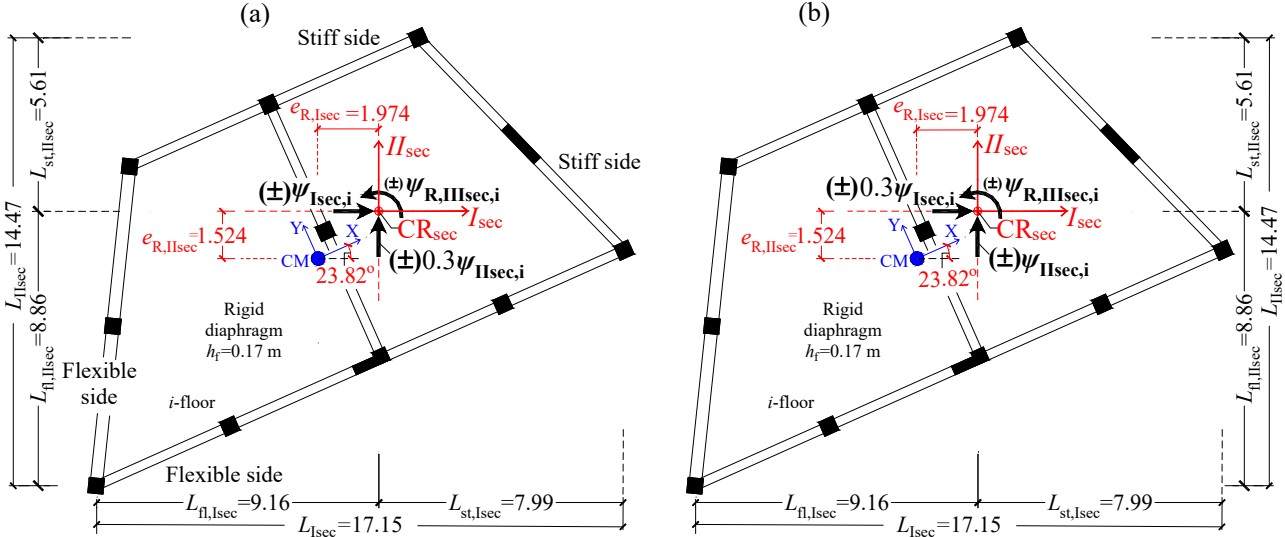

**Figure 10.** Earthquake spatial action of concurrently acting floor enforced-displacements in the framework of 16 pushover analyses: (**a**) 8 combinations $\psi_{Isec,i} \pm 0.3\psi_{IIsec,i} \pm \psi_{R,III\,sec,i}$, to maximize the displacement along $I_{sec}$ axis, (**b**) 8 combinations $0.3\psi_{Isec,i} \pm \psi_{IIsec,i} \pm \psi_{R,III\,sec,i}$, to maximize the displacement along $II_{sec}$ axis.

In other words, the floor enforced displacement vector of this pushover procedure is also applied relative to the "Capable Near Collapse Principal System, $III_{sec}(CR_{sec})$, $I_{sec}$, $II_{sec}$" of the multi-story RC building. The appropriate floor enforced displacements are given in this procedure by graphs or tables following an extended parametric analysis of asymmetric ductile multi-story RC buildings, which is mentioned in Sections 2.3, 2.6 and 2.8. The translational components of the enforced displacements are determined by proposed values of the floor angular deformations at the in-plan position of the vertical ideal principal axis $III_{sec}$. These values are adjusted better to the examined building by performing two sets of temporary pushover analyses. In these analyses, the floor lateral static forces act at the in-plan position of the vertical $III_{sec}$ axis and along the horizontal $I_{sec}$ and $II_{sec}$ axes following the two patterns of Figure 5 and with target displacement given by Table 1. The sixteen (16) simultaneous combinations of the floor enforced displacements are given by tables: 8 combinations $\psi_{Isec,i} \pm 0.3\psi_{IIsec,i} \pm \psi_{R,III\,sec,i}$ and 8 combinations $0.3\psi_{Isec,i} \pm \psi_{IIsec,i} \pm \psi_{R,III\,sec,i}$, where $\psi_{Isec,i}$ and $\psi_{IIsec,i}$ are the translational components and $\psi_{R,III\,sec,i}$ is the rotational one inside each $i$-floor. The first 8 combinations maximize the displacements along the $I_{sec}$ axis, while the second 8 combinations maximize the displacements along the $II_{sec}$ axis.

### 3.8. Analysis Results

The plan inelastic displacement profiles resulted from the proposed pushover procedure compare with the seismic demand ones produced by N-LRHA in Figures 11 and 12 for the building cases 1 and 2, respectively. It is recalled that both the building cases are classified as torsionally sensitive ones according to Equation (9). However, while in case 1 the building is torsionally sensitive at limit, case 2 refers to a more torsionally sensitive building that is well below the limit (1.10). It can be observed that the proposed procedure provides safe floor displacement results. The displacements at the flexible sides, along the $I_{\mathrm{sec}}$ and $II_{sec}$ axis, are estimated conservatively on average by 20% for both the building cases. The displacements at the stiff sides are also conservative on average by 40% and 18% for case 1 and by 24% and 18% for case 2, along the $II_{sec}$ and $I_{sec}$ axis, respectively. The displacements at the in-plan position of the vertical axis $III_{sec}$ ($CR_{\mathrm{sec}}$) and CM are estimated in all floors marginally.

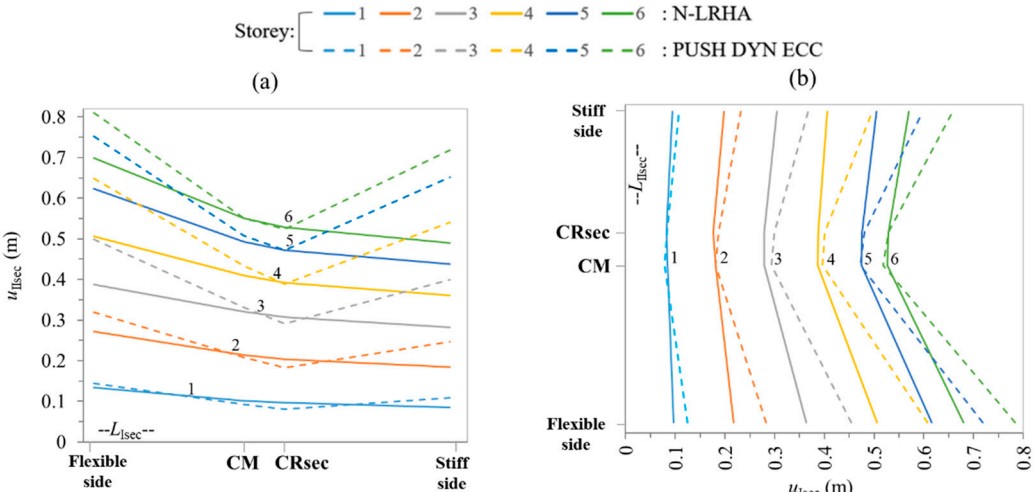

**Figure 11.** Plan inelastic displacement profiles of case 1 building at NC: (**a**) $u_{II\,\mathrm{sec}}$ along the $II_{sec}$ axis, (**b**) $u_{I\,\mathrm{sec}}$ along the $I_{sec}$ axis. Proposed pushover procedure vs. N-LRHA.

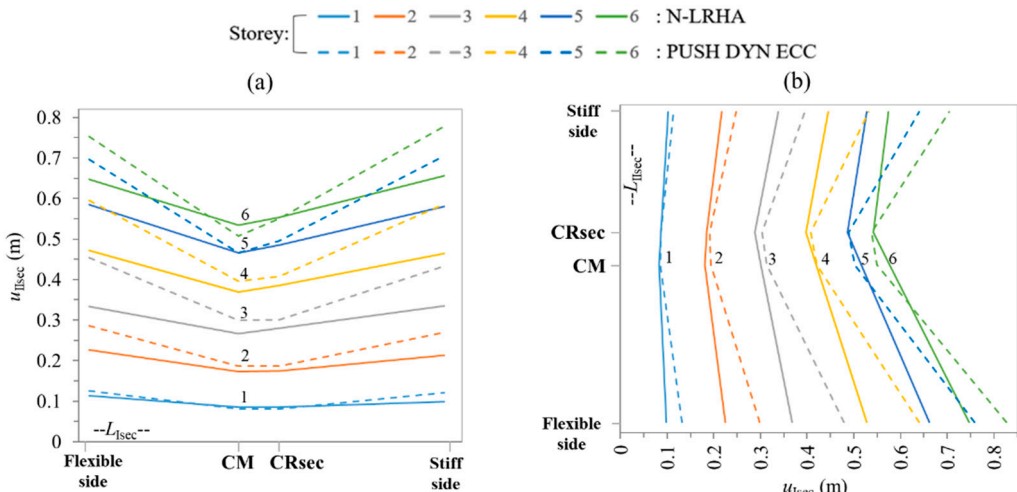

**Figure 12.** Plan inelastic displacement profiles of case 2 building at NC: (**a**) $u_{II\,\mathrm{sec}}$ along the $II_{sec}$ axis, (**b**) $u_{I\,\mathrm{sec}}$ along the $I_{sec}$ axis. Proposed pushover procedure vs. N-LRHA.

In Figures 13 and 14, the floor angular deformations resulted from the proposed pushover procedure compare with the corresponding seismic ones produced by N-LRHA, for the building cases 1 and 2, respectively. Conservative results on average by 4% and 11%

for case 1 and by 10% and 19% for case 2 are observed at the flexible sides along the $II_{sec}$ and $I_{sec}$ axis, respectively. Similarly, conservative results are also observed at the stiff sides, on average by 40% and 22% for case 1 and by 24% and 19% for case 2 along the $II_{sec}$ and $I_{sec}$ axis, respectively.

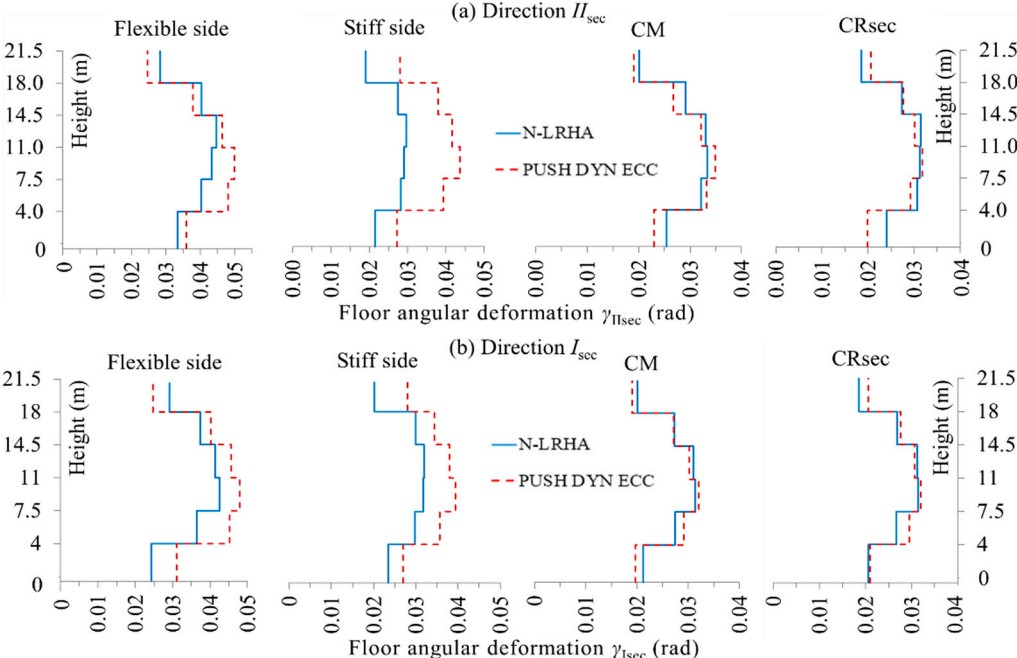

**Figure 13.** Floor angular deformations (rad) of case 1 building at NC: (**a**) $\gamma_{II\,sec}$ along the $II_{sec}$ axis, (**b**) $\gamma_{I\,sec}$ along the $I_{sec}$ axis. Proposed pushover procedure vs. N-LRHA.

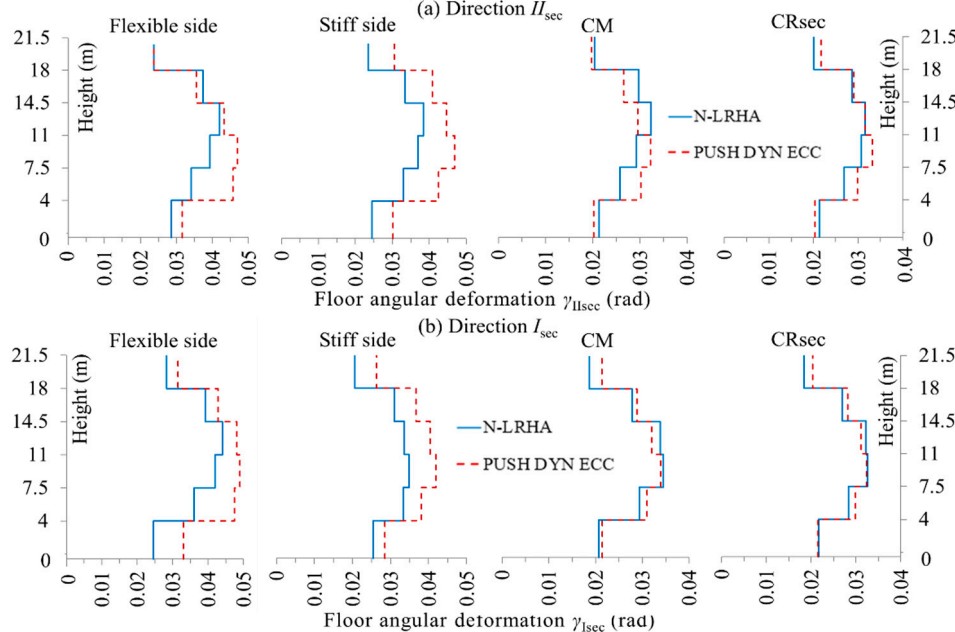

**Figure 14.** Floor angular deformations (rad) of case 2 building at NC: (**a**) $\gamma_{II\,sec}$ along the $II_{sec}$ axis, (**b**) $\gamma_{I\,sec}$ along the $I_{sec}$ axis. Proposed pushover procedure vs. N-LRHA.

Additionally, Figure 15 for case 1 and Figure 16 for case 2 present a detailed comparison of the floor angular deformations resulted by the various pushover procedures (described in detail in Section 3.7) with the seismic demand ones (N-LRHA). The (%) error committed on the estimation of the floor angular deformations at the flexible and stiff sides is recorded

in Tables 4 and 5 for case 1 and in Tables 6 and 7 for case 2, where the negative sign shows non-conservative results for the pushover cases.

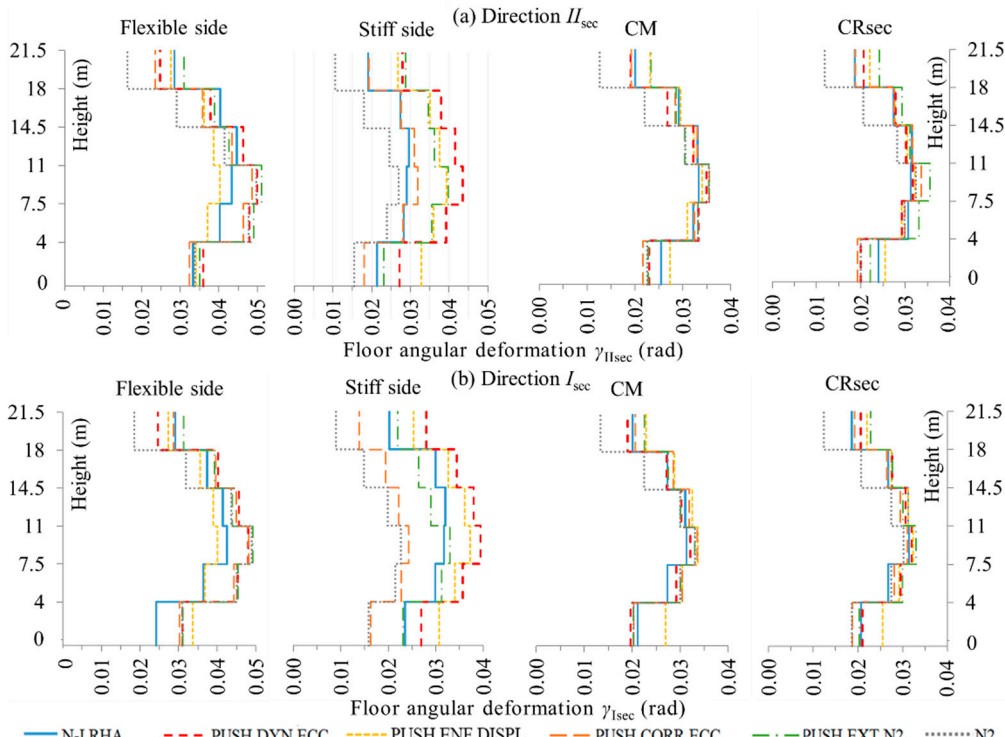

**Figure 15.** Comparison of floor angular deformations at NC resulted from the examined pushover procedures and N-LRHA for case 1: (**a**) $\gamma_{II\,sec}$ along the $II_{sec}$ axis, (**b**) $\gamma_{I\,sec}$ along the $I_{sec}$ axis.

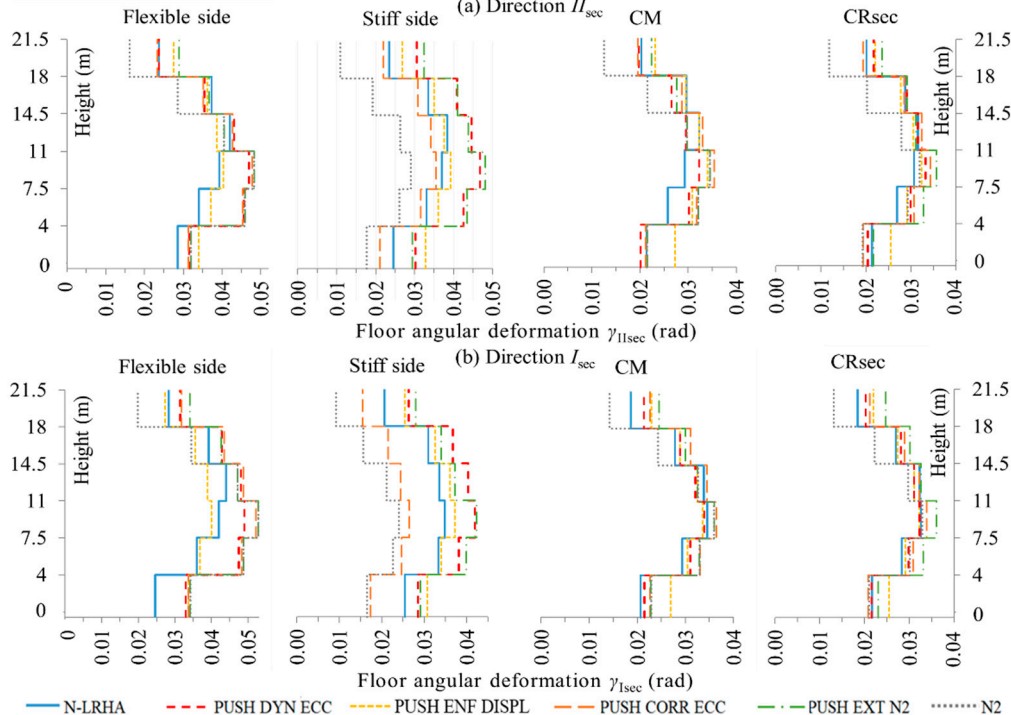

**Figure 16.** Comparison of floor angular deformations at NC resulted from the examined pushover procedures and N-LRHA for case 2: (**a**) $\gamma_{II\,sec}$ along the $II_{sec}$ axis, (**b**) $\gamma_{I\,sec}$ along the $I_{sec}$ axis.

**Table 4.** Error (%) on the seismic floor angular deformations $\gamma_{II\,sec}$ at the stiff and flexible sides along the $II_{sec}$ axis resulted from the examined pushover procedures for case 1 building.

| | Pushover Procedures | | | | | | | | | |
|---|---|---|---|---|---|---|---|---|---|---|
| | Inel. Dynamic Ecc. | | Enforced Displ. | | Corrective Ecc. | | EXT N2 | | N2 | |
| Story | Flexible | Stiff | Flexible | Stiff | Flexible | Stiff | Flexible | Stiff | Flexible | Stiff |
| 1 | 8 | 27 | 2 | 54 | −3 | −16 | 5 | 8 | 1 | −28 |
| 2 | 19 | 39 | −8 | 28 | 15 | −1 | 22 | 26 | 19 | −16 |
| 3 | 15 | 50 | −7 | 35 | 12 | 10 | 18 | 37 | 15 | −7 |
| 4 | 4 | 40 | −14 | 27 | −3 | 4 | −5 | 22 | −7 | −17 |
| 5 | −6 | 38 | −10 | 28 | −11 | 1 | −4 | 26 | −28 | −34 |
| 6 | −13 | 46 | −3 | 40 | −17 | 2 | 9 | 50 | −43 | −45 |

**Table 5.** Error (%) on the seismic floor angular deformations $\gamma_{I\,sec}$ at the stiff and flexible sides along the $I_{sec}$ axis resulted from the examined pushover procedures for case 1 building.

| | Pushover Procedures | | | | | | | | | |
|---|---|---|---|---|---|---|---|---|---|---|
| | Inel. Dynamic Ecc. | | Enforced Displ. | | Corrective Ecc. | | EXT N2 | | N2 | |
| Story | Flexible | Stiff | Flexible | Stiff | Flexible | Stiff | Flexible | Stiff | Flexible | Stiff |
| 1 | 28 | 15 | 39 | 31 | 25 | −31 | 28 | −2 | 28 | −32 |
| 2 | 25 | 20 | 1 | 14 | 22 | −24 | 25 | 5 | 24 | −28 |
| 3 | 13 | 24 | −6 | 17 | 13 | −24 | 16 | 4 | 15 | −29 |
| 4 | 10 | 19 | −6 | 13 | 8 | −30 | 6 | −10 | 5 | −38 |
| 5 | 8 | 15 | −5 | 9 | 6 | −35 | 5 | −12 | −15 | −50 |
| 6 | −15 | 39 | −6 | 26 | −2 | −32 | 7 | 9 | −37 | −56 |

Note: Proposed pushover: Inel. Dynamic Ecc., Enforced Displ. [9], Corrective Ecc. [26], EXT N2 [19], N2 [45].

**Table 6.** Error (%) on the seismic floor angular deformations $\gamma_{II\,sec}$ at the stiff and flexible sides along the $II_{sec}$ axis resulted from the examined pushover procedures for case 2 building.

| | Pushover Procedures | | | | | | | | | |
|---|---|---|---|---|---|---|---|---|---|---|
| | Inel. Dynamic Ecc. | | Enforced Displ. | | Corrective Ecc. | | EXT N2 | | N2 | |
| Story | Flexible | Stiff | Flexible | Stiff | Flexible | Stiff | Flexible | Stiff | Flexible | Stiff |
| 1 | 11 | 23 | 19 | 34 | 10 | −14 | 12 | 20 | 12 | −28 |
| 2 | 34 | 29 | 9 | 9 | 33 | −4 | 35 | 32 | 35 | −21 |
| 3 | 20 | 27 | 3 | 6 | 22 | −4 | 23 | 30 | 23 | −22 |
| 4 | 3 | 16 | −8 | −2 | 2 | −11 | −3 | 14 | −3 | −31 |
| 5 | −5 | 22 | −3 | 5 | −6 | −8 | −2 | 22 | −23 | −43 |
| 6 | 0 | 30 | 16 | 14 | −2 | −7 | 22 | 38 | −32 | −53 |

**Table 7.** Error (%) on the seismic floor angular deformations $\gamma_{I\,sec}$ at the stiff and flexible sides along the $I_{sec}$ axis resulted from the examined pushover procedures for case 2 building.

| | Pushover Procedures | | | | | | | | | |
|---|---|---|---|---|---|---|---|---|---|---|
| | Inel. Dynamic Ecc. | | Enforced Displ. | | Corrective Ecc. | | EXT N2 | | N2 | |
| Story | Flexible | Stiff | Flexible | Stiff | Flexible | Stiff | Flexible | Stiff | Flexible | Stiff |
| 1 | 34 | 12 | 37 | 21 | 37 | −32 | 39 | 14 | 39 | −35 |
| 2 | 32 | 14 | 2 | 2 | 34 | −26 | 35 | 20 | 35 | −32 |
| 3 | 17 | 20 | −5 | 7 | 24 | −24 | 26 | 21 | 26 | −31 |
| 4 | 9 | 20 | −12 | 8 | 11 | −27 | 7 | 11 | 7 | −37 |
| 5 | 9 | 19 | −9 | 5 | 11 | −31 | 9 | 10 | −12 | −49 |
| 6 | 11 | 28 | −4 | 23 | 13 | −25 | 20 | 36 | −30 | −55 |

For the building case 1, which concerns a less torsionally sensitive system, we observe that the N2 pushover procedure seriously underestimates the floor angular deformations at the stiff sides, especially those along the axis $I_{sec}$. The main reason for this is the application of the floor lateral static forces at CM, without the use of any eccentricity. N2 pushover also underestimates the floor angular deformations of the upper floors at all in-plan positions due to the lack of a suitable load pattern that considers the higher-mode effects. The "corrective eccentricities" pushover improves the prediction of the N2 procedure as regards the floor angular deformations at the stiff side along the $II_{sec}$ axis but still underestimates them along the $I_{sec}$ axis. This is due to the low corrective eccentricity used for the estimation of the ductility demands at the stiff sides. The extended N2 pushover provides marginal estimates for the floor angular deformations at the stiff sides along the $I_{sec}$ axis and quite conservative ones along the $II_{sec}$ axis. The "inelastic dynamic eccentricities" and the "enforced displacements" pushovers also provide safe estimates for the floor angular deformations at the stiff sides, which are quite conservative along the $II_{sec}$ axis and balanced along the $I_{sec}$ axis. Additionally, we observe that all pushover procedures provide conservative or marginal estimates for the floor angular deformations at the flexible sides except the N2 pushover which underestimates them at the upper floors of the building.

Similar remarks can be made on the building case 2, which concerns a more torsionally sensitive system. As regards the floor angular deformations at the stiff sides, they are conservatively evaluated by the "inelastic dynamic eccentricities" and the "enforced displacements" pushover procedures, where the latter provides more balanced estimates. Similar conservative estimates are obtained from the extended N2 procedure. In contrast, the N2 procedure seriously underestimates the floor angular deformations at the stiff sides along both the principal directions, more than in case 1, while the "corrective eccentricities" procedure again cannot compensate for the unsafe results along the $I_{sec}$ axis due to the low corrective eccentricity used. As regards the floor angular deformations at the flexible sides, we observe that all pushover procedures provide conservative or marginal estimates except the N2 procedure which underestimates them at the upper floors of the building. Like case 1, the N2 procedure provides unsafe estimates for the floor angular deformations throughout the upper floors due to the lack of a suitable load pattern that considers the higher-mode effects.

As regards the maximum story shears, which are presented indicatively for the building case 2, the pushover procedure with inelastic dynamic eccentricities provides non-conservative results on average by 12% and 20% along the axis $I_{sec}$ and $II_{sec}$, respectively (Figure 17). The maximum base shears shown in Figure 17 are also recorded in the capacity curves of the case 2 building.

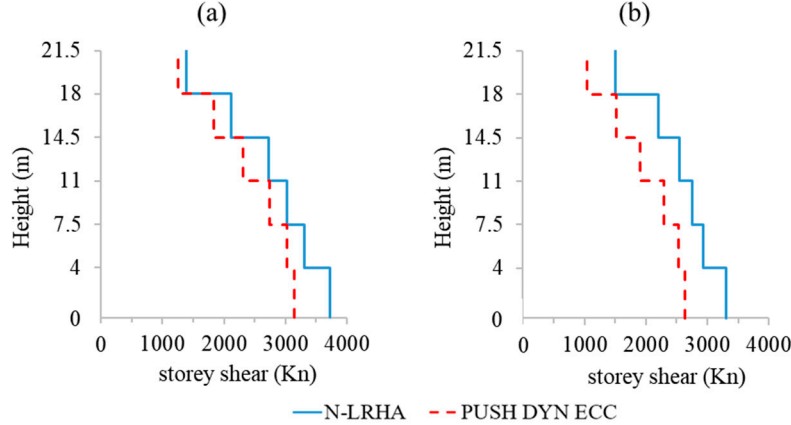

**Figure 17.** Maximum story shears resulted from the "inelastic dynamic eccentricity" pushover procedure on the case 2 building: (**a**) $V_{I \, sec}$ along the $I_{sec}$ axis, (**b**) $V_{II sec}$ along the $II_{sec}$ axis. Comparison with the seismic demand from N-LRHA.

In Figure 18, the mean plastic chord rotations (rad) of the beams' end-sections (for positive and negative bending) at the flexible and stiff sides of the building case 2 resulted by the pushover procedure using inelastic dynamic eccentricities compare with the corresponding ones produced by N-LRHA. We observe that in general the plastic chord rotations of the beams are estimated conservatively on average by 10% and 15% in elevation. The (%) error on the estimation of the mean plastic chord rotations of the beams' end-sections, at the flexible and stiff sides of each floor, by the proposed pushover procedure is presented in Table 8, indicatively for the case 2 building.

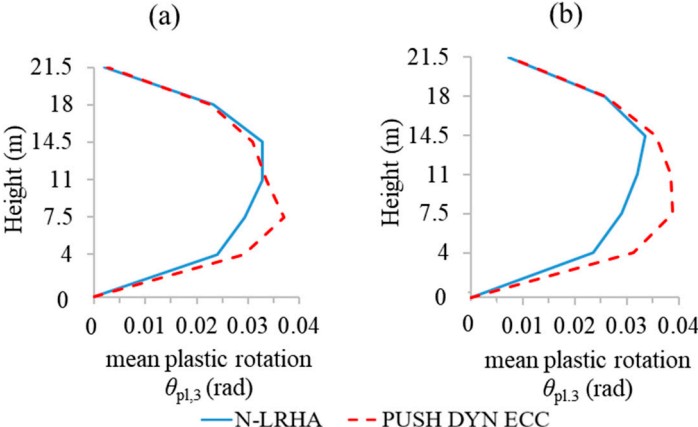

**Figure 18.** Mean plastic chord rotations $\theta_{pl,3}$ (rad) of the beams' end-sections at: (**a**) the flexible sides and (**b**) the stiff sides of the case 2 building resulted by the "inelastic dynamic eccentricities" pushover procedure at the NC state.

**Table 8.** Error (%) on the mean seismic plastic chord rotations of the beams' end-sections at the flexible and stiff sides of the case 2 building resulted by the "inelastic dynamic eccentricities" pushover procedure at the NC state.

| Story | Flexible Sides (%) | Stiff Sides (%) |
|:-----:|:------------------:|:---------------:|
| 1 | 22 | 33 |
| 2 | 26 | 34 |
| 3 | 3 | 20 |
| 4 | −6 | 6 |
| 5 | −2 | 1 |
| 6 | 29 | 4 |

Additionally, the (%) error on the estimation of the mean plastic chord rotations $\theta_{pl,3}$ and $\theta_{pl,2}$ of the columns' and walls' base sections are presented in Table 9 indicatively for the case 2 building, where the subscripts 3 and 2 denote the local axes of a base column/wall section normal to its strong and weak direction, respectively. We observe that the proposed pushover procedure provides in general balanced estimates of the plastic chord rotations of the vertical elements, which are non-conservative for the weak direction of the wall.

**Table 9.** Error (%) on the mean seismic plastic chord rotations $\theta_{pl,3}$ and $\theta_{pl,2}$ of the base sections of columns and walls of the case 2 building resulted by the "inelastic dynamic eccentricities" pushover procedure at the NC state.

| Base Sections | $\theta_{pl,2}$ (%) | $\theta_{pl,3}$ (%) |
|:-------------:|:-------------------:|:-------------------:|
| Columns | 23 | −7 |
| Walls | −15 | 22 |

The capacity curves resulted from the "inelastic dynamic eccentricities" pushover procedure on the case 2 building are shown in Figure 19b, indicatively for the eight

separate pushovers along the $I_{sec}$ and $II_{sec}$ axes that use the first modal pattern of lateral forces. Figure 19a also shows the capacity curves from the application of the N2 pushover procedure. The curves have been idealized with the bi-linear technique and the top displacement of the building at yield as well as the corresponding maximum base shear, for the various in-plan positions of the external floor lateral loading along the $I_{sec}$ or $II_{sec}$ axis (Figure 7), are marked in Figure 19. The maximum base shears are about equal to the values shown in Figure 17. Loading towards the stiff sides (load cases 3,4 and 7,8 in Figures 7 and 19b) shows higher secant stiffness $K_{sec}$ at yield while the opposite happens for loading towards the flexible sides of the building (cases 1,2 and 5,6 in Figures 7 and 19b). It is emphasized that the first (elastic) line of the bi-linear diagrams, leading directly to the yield point of the building, follows the initial constant slope of the capacity curves. This treatment is thereby consistent with the global use of the secant stiffness $EI_{sec}$ at yield in all structural elements, which is the lowest value of effective elastic stiffness.

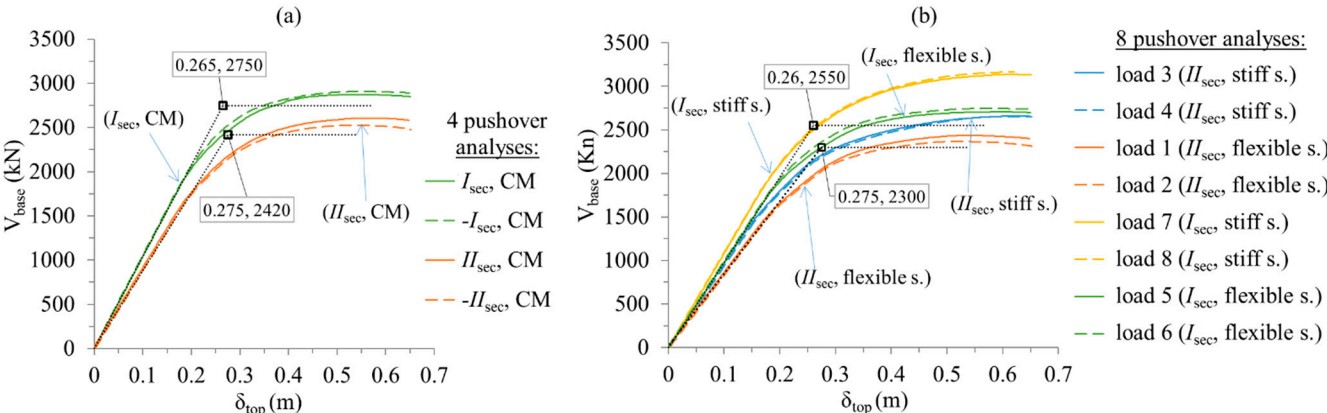

**Figure 19.** Capacity curves of the case 2 building resulted from: (**a**) N2 pushover procedure (EN 1998-1), (**b**) "inelastic dynamic eccentricities" pushover procedure. Load numbering is shown in Figure 7.

Finally, the 6-story building of case 2 (that with the increased floor moment of inertia) will be verified at the SD state (Section 2.8), which is reached when D/C ratios in terms of chord rotations of the structural members at ultimate state are equal to 0.75 ($\theta_{SD} \leq 0.75\theta_{um}$) according to EN 1998-1. For this purpose, the effective bending stiffness of the structural members of the nonlinear model appropriate for the SD state is considered equal to the average of the corresponding ones used for the verification at the NC state and those that considered appropriate for the DL state. The former is the secant stiffness $EI_{sec}$ at yield while the latter is taken equal to $EI_{eff} = 2EI_{sec}$, i.e., double of that at the NC state, but not less than the 25% and also not greater than the 50% of the geometrical stiffness. The mean values of the ratio of the effective stiffness to the corresponding geometric one of the structural elements ($EI_{eff}/EI_g$) in each floor are about equal 1.5 times of the reported ones in Table 2 for the verification of the building at the NC state. Then, the "ideal effective principal system" of the building, at the SD state, is defined according to the process of Section 2.2. The inelastic static eccentricities and the direction of the horizontal ideal principal axes are close enough to the corresponding values of the nonlinear model appropriate for the verification of the NC state (Section 3.3). The building is characterized as torsionally sensitive according to Equation (9), because the smaller of the two torsional radii ratios (0.80) is again lower than 1.10. The inelastic dynamic eccentricities are calculated again by Equations (10) and (11) for torsionally sensitive buildings, as in Section 3.4, and the floor lateral forces are applied using them (Figure 7) following the two proposed modal patterns, where in the second one an additional lateral force (equal to the 20% of the distributed base shear) is applied at the top floor. The floor lateral load patterns have approximately the shapes shown in Figure 8 which are determined by an uncoupled modal analysis. N-LRHA is performed again using the three pairs of five accelerograms shown in Figure 9, which are set to a PGA equal to 0.30g that causes the SD state of the building.

The seismic displacement at the position of the vertical ideal effective principal axis in the top floor is equal to 0.33 m, which is less than the corresponding prediction obtained from Table 1 reduced by 30% according to Section 2.8 ($0.7 \cdot 0.52 = 0.36$ m). The eight (8) separate pushovers (per pattern) with inelastic dynamic eccentricities are performed with target displacement equal to that of the N-LRHA and the envelope of the thirty-two (32) combined effects by the SRSS rule of the sixteen (16) total pushovers (eight per pattern) is considered as representative of the spatial seismic action at the SD state, according to Section 2.7.

In Figures 20 and 21, the plan inelastic displacement profiles and the floor angular deformations in elevation at the SD state are presented. We observe that the floor displacements at the flexible and stiff sides of the building are conservatively estimated in all floors, on average by 10% and 15% and 11% and 14%, respectively, along the horizontal ideal principal axes $II_{eff}$ and $I_{eff}$. Similar conservative estimates are obtained for the floor angular deformations at the flexible and stiff sides of the building along both the principal axes, on average by 12 and 15%, respectively.

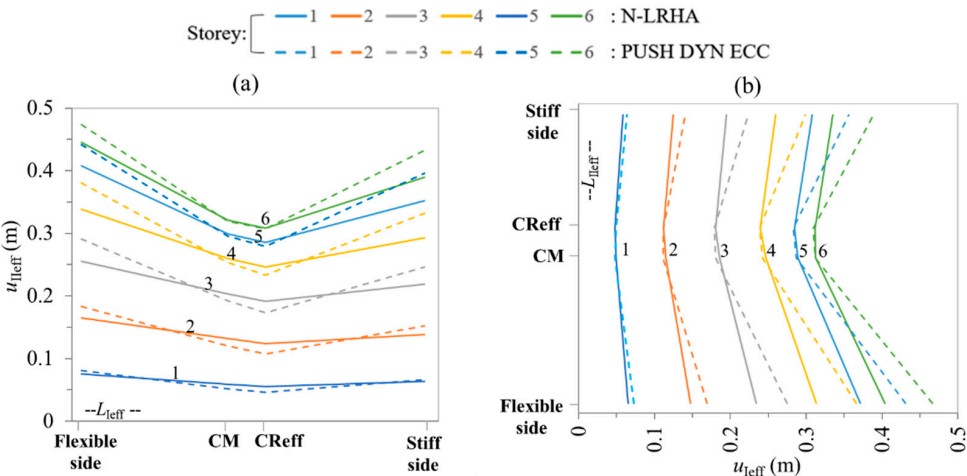

**Figure 20.** Plan inelastic displacement profiles of case 2 building at SD: (**a**) $u_{IIeff}$ along the $II_{eff}$ axis, (**b**) $u_{Ieff}$ along the $I_{eff}$ axis. Proposed pushover procedure vs. N-LRHA.

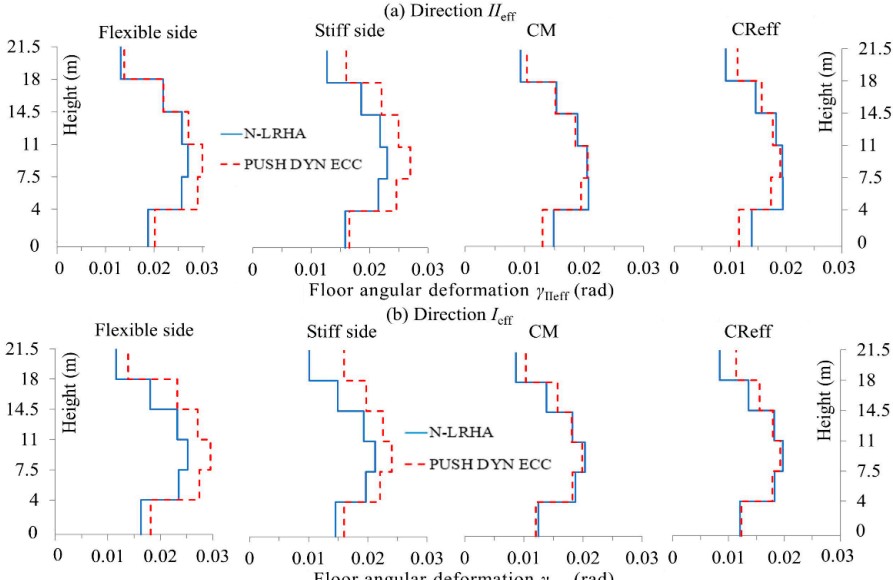

**Figure 21.** Floor angular deformations (rad) of case 2 building at SD: (**a**) $\gamma_{IIeff}$ along the $II_{eff}$ axis, (**b**) $\gamma_{Ieff}$ along the $I_{eff}$ axis. Proposed pushover procedure vs. N-LRHA.

## 4. Conclusions

A recently proposed procedure of non-linear static (pushover) analysis on single-story RC buildings using inelastic dynamic eccentricities is extended and appropriately refined in the current paper with a view to serve as a rational seismic assessment tool of multi-story RC buildings at the Near Collapse (NC) state.

The new proposed pushover procedure on multi-story RC building uses the already defined inelastic dynamic eccentricities for single-story RC buildings [4] to apply the lateral loads at two different positions on each floor, in order to take account of the influence of the coupled torsional/translational response on the ductility demands at the stiff and flexible sides. Consideration of the accidental eccentricities can also be obtained by using the inelastic design eccentricities, which combine the former ones with the inelastic dynamic eccentricities. The new procedure on multi-story RC buildings differs from the corresponding one for single-story buildings in the following points: (a) the definition of the "Capable Near Collapse Principal System $CR_{sec}(III_{sec})$, $I_{sec}$, $II_{sec}$" of the multi-story building using the well-known methodology of the torsional optimum axis, which is the reference inelastic system for the application of the pushover procedure, (b) the definition of the corresponding "Capable Near Collapse Torsional Radii, $(r_{I,sec}, r_{II,sec})$" of the multi-story building, in order to classify the building as torsionally sensitive or not and (c) the use of two patterns of floor lateral static forces in elevation, both proportional to the uncoupled translational mode-shapes but the second one with an additional top force, to take account of the higher-mode effects.

Using the inelastic dynamic or design eccentricities (in-plan) and two patterns of lateral static forces (in elevation), the safe estimation of the ductility demands at the stiff and flexible sides of the building as well as those at the higher floors throughout the building at the NC state is achieved by the envelope of 32 SRRS combinations (16 per pattern) of the effects of sixteen separate pushover analyses (eight per pattern). The proposed "inelastic dynamic eccentricities" procedure can also be used for the verification of multi-story RC buildings at the SD and DL state. For this purpose, appropriate modification of the lateral stiffness of the building and of the target displacement is proposed.

The effectiveness of the proposed pushover procedure is verified on a six-story, double asymmetric, torsionally sensitive, RC building, relative to the results of N-LRHA. The evaluation is made in terms of floor displacements, floor angular deformations, story shears and plastic chord rotations. An additional comparison is made with the N2 (EN 1998-1), the extended N2 [19], the "corrective eccentricities [26] and the "enforced displacements" [9] pushover procedures, in terms of floor angular deformations. Two cases of torsional sensitivity were examined. In both cases the building is classified as torsionally sensitive according to the proposed procedure, in the first case at the limit while in the second one well below the limit. On the contrary, only the second case building is classified as torsionally sensitive according to EN 1998-1 while the first one is about torsionally non-sensitive buildings. The key findings of the current investigation and the main conclusions are:

(a) The N2 (EN 1998-1) pushover procedure seriously underestimates the floor displacements and the floor angular deformations at the stiff sides. This was recorded for both the examined cases of torsional sensitivity, where the first one refers to a torsionally non-sensitive building according to EN 1998-1. Additionally, it underestimates the seismic demand at the higher floors throughout the building. Therefore, inelastic dynamic eccentricities and appropriate loading patterns must be used in the framework of pushover analysis.

(b) The extended N2 pushover procedure corrects the unsafe estimates of the N2 procedure and provides in general conservative estimates of the floor angular deformations throughout the building.

(c) The "corrective eccentricities" pushover procedure, with the use of the two modal patterns of the floor lateral static forces proposed herein, also corrects the unsafe results of the N2 procedure at the higher floors but it still provides in general unsafe

estimates of the floor angular deformations at the stiff sides of the building due to the small value of the corrective eccentricity.

(d) The "enforced displacements" pushover procedure, a recently developed method, provides in general conservative or marginal estimates of the floor angular deformations throughout the building, as shown in Tables 4–7.

(e) The "inelastic dynamic eccentricities" pushover procedure on multi-story RC buildings provides in general safe results for the floor displacements and the floor angular deformations at the stiff and flexible sides as well as for those at the higher floors, at the NC state. Wherever unconservative values are shown, they are just below the seismic demand. Additionally, the conservatism of the proposed procedure is not higher than in other examined pushover procedures. The maximum story shears are a little underestimated by the proposed pushover procedure. The plastic chord rotations of the end-sections of beams and columns/walls are in general predicted with safety. The developed plastic mechanism of the building at the NC state is also conservatively assessed by the proposed procedure. Additionally, the proposed procedure provides conservative estimates of the seismic demand for the verification at the SD state, close enough to the predictions of N-LRHA.

Consequently, the proposed "inelastic dynamic eccentricities" pushover procedure is an effective tool for the seismic assessment of multi-story RC buildings. It safely predicts the coupled torsional/translational response and the higher-mode effects and sufficiently improves the unsafe estimations of the conventional pushover analysis while retaining the simplicity of the original procedure proposed for single-story buildings. The procedure is promising; however, a broader investigation should be performed in order to state a general conclusion.

**Author Contributions:** Conceptualization, A.B., T.M. and A.A.; methodology, A.B., T.M. and A.A.; software, A.B., T.M. and A.A.; validation, A.B., T.M. and A.A.; formal analysis, A.B., T.M. and A.A.; investigation, A.B., T.M. and A.A.; resources, A.B., T.M. and A.A.; data curation, A.B., T.M. and A.A.; writing—original draft preparation, A.B.; writing—review and editing, A.B., T.M. and A.A.; visualization, A.B., T.M. and A.A.; supervision, T.M. and A.A.; project administration, T.M. and A.A.; funding acquisition, None. All authors have read and agreed to the published version of the manuscript.

**Funding:** This research received no external funding.

**Institutional Review Board Statement:** Not applicable.

**Informed Consent Statement:** Not applicable.

**Data Availability Statement:** The data presented in this study are available in the article.

**Conflicts of Interest:** The authors declare no conflict of interest.

### Appendix A

The section geometry of the structural elements as well as the reinforcement details of the columns/walls and beams of the six-story building of Figure 6 are presented here in Tables A1–A3, respectively. The numbering of the structural members in the analysis model of the building is shown in Figure A1.

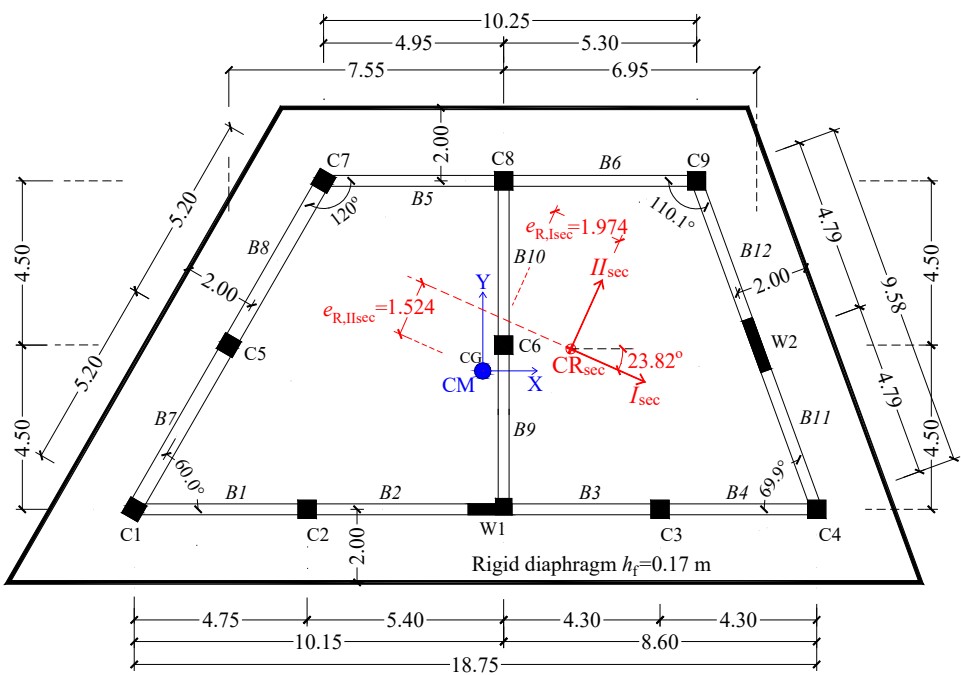

**Figure A1.** Numbering of structural members in the analysis model of the six-story building.

**Table A1.** Member sections of the six-story building (in cm).

| Structural Element | Story | | | | | |
|---|---|---|---|---|---|---|
| | 1 | 2 | 3 | 4 | 5 | 6 |
| All Columns | 55/55 | | | 50/50 | | |
| Wall W1 | 120/30/45/45 | | | | | |
| Wall W2 | 30/150 | | | | | |
| Beams 1–6 and 9–12 | T 30/60/150/17 | | | T 30/55/150/17 | | |
| Beams 7–8 | T 40/60/160/17 | | | T 40/55/160/17 | | |

**Table A2.** Longitudinal and transverse confinement reinforcement (hoops) details of columns/walls of the six-story building.

| Columns, Walls | Story | | | | | |
|---|---|---|---|---|---|---|
| | 1 | 2 | 3 | 4 | 5 | 6 |
| C 1 | 16Ø20 hoops, 5 ties Ø8/84 | | 12Ø20 hoops, 4 ties Ø10/80 | | | |
| C 5 | 16Ø20 hoops, 5 ties Ø8/84 | | 12Ø20 hoops, 4 ties Ø10/80 | | 12Ø20 hoops, 4 ties Ø8/84 | 4Ø20+8Ø14 hoops, 4 ties Ø8/84 |
| C 2–4, 6–9 | 12Ø20 hoops, 4 ties Ø10/80 | | | | | |
| W 1 | 10Ø20 + 12Ø20(Column) + 6Ø10 hoops, 4 ties Ø10/80 | | (2Ø20 + 8Ø14) + (4Ø20+8Ø14) (Column) + 6Ø10 hoops, 4 ties Ø8/100 + 4Ø8/84 | | | |
| W 2 | 2 × (10Ø 20) + 12Ø10 hoops, 4 ties Ø10/80 | | 2 × (2Ø20 + 8Ø16) + 12Ø10 hoops, 4 ties Ø8/100 | | | |

Note: In walls, the longitudinal reinforcement is reported at the two edges (wall-columns inside the width) and at the web (between the two edges), e.g., 2 × (10Ø 20) + 12Ø10. All hoops are closed according to EN 1998-1 and the placement distances are in mm, e.g., hoops, 4 ties Ø8/100.

**Table A3.** Longitudinal reinforcement details of beams of the six-story building. All beams have steel bars of 16mm diameter, and the number of bars is reported at the upper and lower fibers of the end-sections, at the start (*s*) and at the end (*e*) of beams (e.g., *s* 8-7 means 8Ø16 at the upper fiber and 7Ø16 at the lower fiber of the start section of the beam).

| Beam | Story | | | | | | | | | | | |
|------|-------|---|---|---|---|---|---|---|---|---|---|---|
| | 1 | | 2 | | 3 | | 4 | | 5 | | 6 | |
| | *s* | *e* | *s* | *e* | *s* | *e* | *s* | *e* | *s* | *e* | *s* | *e* |
| | Number of Longitudinal Steel Bars D = 16 mm in the Upper and Lower Fibers at the Start and End Sections of Beams | | | | | | | | | | | |
| B 1 | 8-7 | 9-7 | 8-7 | 9-7 | 7-6 | 8-6 | 6-5 | 7-5 | 5-4 | 5-4 | 4-4 | 4-4 |
| 2 | 9-7 | 10-8 | 9-7 | 10-8 | 8-6 | 9-7 | 7-5 | 8-6 | 5-4 | 6-5 | 4-4 | 4-4 |
| 3 | 10-8 | 9-7 | 10-8 | 9-7 | 9-7 | 8-6 | 8-6 | 6-5 | 6-5 | 5-4 | 4-4 | 4-4 |
| 4 | 9-7 | 9-8 | 9-7 | 9-8 | 8-6 | 7-6 | 6-5 | 6-5 | 5-4 | 5-4 | 4-4 | 4-4 |
| 5 | 8-6 | 8-6 | 8-6 | 7-5 | 7-5 | 6-5 | 6-4 | 5-4 | 5-4 | 4-4 | 4-4 | 4-4 |
| 6 | 8-6 | 8-6 | 7-5 | 8-6 | 6-5 | 7-5 | 5-4 | 6-4 | 4-4 | 5-4 | 4-4 | 4-4 |
| 7 | 11-9 | 10-7 | 11-8 | 10-7 | 10-7 | 9-7 | 8-6 | 7-5 | 6-4 | 6-4 | 5-5 | 5-5 |
| 8 | 10-7 | 11-9 | 10-7 | 11-8 | 9-7 | 10-7 | 7-5 | 8-6 | 6-4 | 6-4 | 5-5 | 5-5 |
| 9 | 5-5 | 7-5 | 5-5 | 7-5 | 5-5 | 6-5 | 4-4 | 5-4 | 4-4 | 4-4 | 4-4 | 4-4 |
| 10 | 7-5 | 7-6 | 7-5 | 7-7 | 6-5 | 6-5 | 5-4 | 5-4 | 4-4 | 4-4 | 4-4 | 4-4 |
| 11 | 8-6 | 9-7 | 8-6 | 9-7 | 7-6 | 8-6 | 6-5 | 7-5 | 4-4 | 4-4 | 4-4 | 4-4 |
| 12 | 9-7 | 8-6 | 9-7 | 8-6 | 8-6 | 7-6 | 7-5 | 6-5 | 4-4 | 4-4 | 4-4 | 4-4 |

Note: At the upper fiber, 12 bars of diameter 8 mm inside the effective slab width (1.5 or 1.6 m) are also accounted for negative bending. The confinement reinforcement at the end-sections of all beams consists of closed hoops with 2 ties of diameter 8 mm placed every 84 mm.

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
