# Peer review of "Inelastic Dynamic Eccentricities in Pushover Analysis Procedure of Multi-Story RC Buildings"

_buildings, doi:10.3390/buildings11050195_

Round 1

Reviewer 1 Report

The paper proposes an extension of a procedure for assessing the seismic performance of irregular buildings  through pushover methods., accounting for higher modes effects. The paper is interesting and well-presented, despite in some parts is very redundant. I suggest to authors of streamlining the text, in order to allow readers of better capturing the proposed concepts. Some comments are provided in the attached PDF file, which can be followed by authors for improving the manuscript, before to consider the paper ready for publication

Author Response

Thank  very much the reviewers for their useful comments

Reviewer 2 Report

Brief summary

The research presented in this paper describes a pushover procedure that takes advantage of the inelastic dynamic eccentricities. This methodology aims to identify and predict the seismic ductility demand, estimate peak interstorey drifts and predict plastic mechanism. To validate the procedure, a six-storey RC building was selected, and the results were compared with nonlinear dynamic analyses. The Authors concluded that the outcomes of the study provide conservative results, and then the procedure can be safely used.

Broad comments

The topic of the study is interesting and worthy of investigation. The paper is well-written and well-organised. The presented research follows previous recent studies adopted for single-storey buildings; this paper applies the same methodologies to a multi-storey building.

The study bases its findings on the full development of plastic hinges at both end-sections of the structural elements. Albeit the building was designed in compliance with the European codes, a worth-noting aspect is the possible occurrence of brittle failure (shear failure, joints failure). Did the Authors conduct such checks?

In order to compare the procedure with nonlinear time-history analyses, the Authors developed five records. However, the European code recommends adopting at least seven accelerograms. The Reviewer suggests considering the possibility of increasing the number of ground motions at least to seven.

Specific comments

In the abstract, the Authors refer to “peak inter-storey drifts”. However, through the manuscript, they often refer to “floor angular deformation”. Albeit both denominations are acceptable, it should be better to be consistent and always refer to the same. The Reviewer suggests replacing “floor angular deformation” with the more common “inter-storey drifts”.

Line 98. For the first time, it is introduced the nomenclature “CRsec(IIIsec),Isec,IIsec”. For a clearer understanding, it could be helpful to the reader to clarify here its meaning.

Line 112. Consider replacing “have already be examined” with “have already been examined”.

Line 132. The assumption that all the plastic hinges have been developed means that, as also underlined by the Authors, a global mechanism has been experienced by the structure. It can be the case of new buildings, but unlikely it can happen in existing buildings. Please add some considerations in this sense.

Line 896. The Reviewer partially agrees with the Authors in assessing that the “inelastic dynamic eccentricities” pushover procedure is an effective tool for the seismic assessment of multi-storey buildings. The procedure is promising, however, a broader investigation should be performed in order to state a general conclusion. Please consider rephrasing the sentence.

Author Response

All comments have replied and are shown  in the text in red color.

See attached file

Reviewer 3 Report

Thank you for this contribution. This is an interesting and timely manuscript. This paper discusses how inelastic dynamic eccentricities in pushover analysis for multi-storey RC buildings. The conducted analysis is typically standard and falls within the expected work from such a publication and hence the work merits publication. As such, the authors are invited to properly address the following items:

  1. The details on the FE models are quite superficial and light. Please ensure that your model is properly described to enable interested researchers from extending and replicating your work. Special attention should be paid to specifics such as element type (DOFs), convergence criteria and performance metrics.
  2. It would be great if the authors can provide a range for dynamic eccentricities that may lead to large possibility of damage.
  3. Nearing a NC state could also be tied to having a probabilistic nature. How such nature is taken into account in the proposed method?
  4. Finally, were the authors able to conduct reliability analysis on the verification of their proposed method, as compared to other methods?

Author Response

Thank  very much the reviewers for their useful comments. All comments have replied and are shown  in the text in red color.

Round 2

Reviewer 1 Report

After the review made by authors, the paper can be accepted for publication 

Author Response

Thank you for your review.

Reviewer 2 Report

Broad comments

The Reviewer appreciates the efforts to revise the manuscript. The Authors have addressed the reviewers' comments adequately.

Regarding the limited number of ground motions adopted, the Authors decided not to increase their number. Although it is pretty known what EN 1998-1 suggests, the Reviewer's recommendation was based on the fact that the article is presenting a research study rather than a practitioner’s design. However, the added sentence in the conclusion paragraph includes somehow this aspect. In the Reviewer’s opinion, only the following comment should be addressed before accepting the paper for publication.

Specific comments

In the new reference 10, please replace "Uciale" with "Ufficiale".

Author Response

(The authors gave the same response as above.)

Reviewer 3 Report

thanks for your efforts.

Author Response

Thank you for your review.